# Optimal Iterative Sketching with the Subsampled Randomized Hadamard Transform

**Jonathan Lacotte**[*]
Department of Electrical Engineering
Stanford University
lacotte@stanford.edu

**Sifan Liu**[*]
Department of Statistics
Stanford University
sfliu@stanford.edu

**Edgar Dobriban**
Department of Statistics
University of Pennsylvania
dobriban@wharton.upenn.edu

**Mert Pilanci**
Department of Electrical Engineering
Stanford University
pilanci@stanford.edu

## Abstract

Random projections or sketching are widely used in many algorithmic and learning contexts. Here we study the performance of iterative Hessian sketch for least-squares problems. By leveraging and extending recent results from random matrix theory on the limiting spectrum of matrices randomly projected with the subsampled randomized Hadamard transform, and truncated Haar matrices, we can study and compare the resulting algorithms to a level of precision that has not been possible before. Our technical contributions include a novel formula for the second moment of the inverse of projected matrices. We also find simple closed-form expressions for asymptotically optimal step-sizes and convergence rates. These show that the convergence rate for Haar and randomized Hadamard matrices are identical, and asymptotically improve upon Gaussian random projections. These techniques may be applied to other algorithms that employ randomized dimension reduction.

## 1 Introduction

Random projections are a classical way of performing dimensionality reduction, and are widely used in many algorithmic and learning contexts, e.g., [32, 17, 35, 9] etc. In this work, we study the performance of the iterative Hessian sketch [24], in the context of overdetermined least-squares problems

$$x^* := \operatorname*{argmin}_{x \in \mathbb{R}^d} \left\{ f(x) := \frac{1}{2} \|Ax - b\|^2 \right\} . \tag{1}$$

Here $A \in \mathbb{R}^{n \times d}$ is a given data matrix with $n \geqslant d$ and $b \in \mathbb{R}^n$ is a vector of observations. For simplicity of notations, we assume throughout this work that $\mathrm{rank}(A) = d$. We will leverage and extend recent results on the limiting spectral distributions of two classical subspace embeddings, random uniform projections and the subsampled randomized Hadamard transform (SRHT), to compare corresponding iterative Hessian sketch versions.

The iterative Hessian sketch (IHS) is an effective iterative method for solving least-squares [23, 24, 14, 28] (and more general convex optimal optimization problems [25]), and it aims to address the condition number dependency of standard iterative solvers as follows. Given step sizes $\{\mu_t\}$ and

---

[*]Equal contributions.

momentum parameters $\{\beta_t\}$, it computes the update

$$x_{t+1} = x_t - \mu_t H_t^{-1} \nabla f(x_t) + \beta_t(x_t - x_{t-1}), \qquad (2)$$

where the Hessian $H = A^\top A$ of $f$ is approximated by $H_t = A^\top S_t^\top S_t A$, and $S_0, \dots, S_t, \dots$ are i.i.d. sketching (random) matrices with dimensions $m \times n$ and $m \ll n$. From now on, we refer to the i.i.d. property of the sketching matrices as *refreshed* matrices.

There are many possible choices for the sketching matrices $S_t$, and this is critical for the performance of the IHS. A classical sketch is a matrix $S \in \mathbb{R}^{m \times n}$ with independent and identically distributed (i.i.d.) Gaussian entries $\mathcal{N}(0, m^{-1})$, for which the matrix multiplication $SA$ requires in general $\mathcal{O}(mnd)$ basic operations (using classical matrix multiplication). This is larger than the cost $\mathcal{O}(nd^2)$ of solving (1) with direct methods when $m \geqslant d$. Another well-studied embedding is the (truncated) $m \times n$ *Haar* matrix $S$, whose rows are orthonormal and with range uniformly distributed among the subspaces of $\mathbb{R}^n$ with dimension $m$. However, this requires time $\mathcal{O}(nm^2)$ to be formed, through a Gram-Schmidt procedure, which is also larger than $\mathcal{O}(nd^2)$.

The SRHT [1, 27] is another classical random orthogonal embedding. Due to the recursive structure of the Hadamard transform, the sketch $SA$ can be formed in $\mathcal{O}(nd \log m)$ time, so that the SRHT is often viewed as a standard reference point for comparing sketching algorithms. Moreover, for many applications, random projections with i.i.d. entries perform worse compared to orthogonal projections [17, 18, 9]. More recently, this observation has also found some theoretical support in limited contexts [8, 36]. Works by [6] also showed the guaranteed improved performance in accuracy and/or speed. Consequently, along with computational considerations, these results favor the SRHT over Gaussian projections.

Our goal in this work is to design an optimal version of the IHS with SRHT and Haar embeddings. For this purpose, it is necessary to have a tight characterization of the spectral properties of the matrix $U^\top S^\top SU$ where $U$ is an $n \times d$ partial orthogonal matrix (see, e.g., [13]). With Gaussian embeddings, the matrix $U^\top S^\top SU$ has the well-studied Wishart distribution, see e.g., [19, 3, 29, 5, 7, 38]. In fact, [13] provided an optimal IHS with Gaussian embeddings, and showed that the best achievable error $\|A(x_t - x^*)\|^2$ scales as $(d/m)^t$. However, a similar analysis does not work for SRHT and Haar sketches. To make progress on this problem, we aim to leverage powerful tools from asymptotic random matrix theory, and *we consider the asymptotic regime where we let the relevant dimensions go to infinity*.

Our technical analysis is based on asymptotic random matrix theory, see e.g., [3, 29, 5, 7, 38] etc. Classical results such as the Marchenko-Pastur law do not address well the case of the SRHT, and we leverage recent results on *asymptotically liberating sequences* established by [2] (see also [31] for prior work). Further, we are inspired by the work of [8], who, to our knowledge, first leveraged these results to study the SRHT. However, their results are limited to one-step "sketch-and-solve" methods, and do not address the iterative Hessian sketch. Moreover, while we build on their results, we also need to extend them significantly: for instance, we need to derive the second moment formula for $\theta_{2,h}$ in (11), which is novel and non-trivial to establish.

Beyond the IHS, there exist other randomized pre-conditioning methods [4, 10, 20, 26] for solving least-squares, which are based on the SRHT (or closely related sketches) which address effectively the condition number dependency of iterative solvers. Besides least-squares, SRHT sketches are widely used for a wide range of applications across numerical linear algebra, statistics and convex optimization, such as low-rank matrix factorization [11, 34], kernel regression [37], random subspace optimization [16], or sketch and solve linear regression [8], see the reviews above for applications. Hence, a refined analysis of the SRHT, including our specific technical contributions, may also lead to better algorithms in these fields.

Throughout the paper, we will consistently use the following assumptions and notations for the aspect ratios, $\gamma := \lim_{n,d \to \infty} \frac{d}{n} \in (0, 1)$, $\xi := \lim_{n,m \to \infty} \frac{m}{n} \in (\gamma, 1)$ and $\rho_g := \frac{\gamma}{\xi} \in (0, 1)$, and the subscript $g$ (resp. $h$) will refer to Gaussian-related (resp. Haar and Hadamard-related) quantities. We use the notations $\|z\| \equiv \|z\|_2$ for the Euclidean norm of a real vector $z$, $\|M\|_2$ for the operator norm of a matrix $M$, and $\|M\|_F$ for its Frobenius norm. For a sequence of iterates $\{x_t\}$, we denote the error vector $\Delta_t := U^\top A(x_t - x^*)$, where $U$ is the $n \times d$ matrix of left singular vectors of $A$. In particular, we have that $\|\Delta_t\|^2 = \|A(x_t - x^*)\|^2$.

## 1.1 Overview of our results, contributions and questions left open

All our contributions hold in the asymptotic limit $n, d, m \to \infty$, and under the aforementioned assumption that the aspect ratios $(d/n)$ and $(m/n)$ have finite limits.

We work with the matrix $U^\top S^\top S U$, where $U$ is an $n \times d$ matrix with orthonormal columns and $S$ is an $m \times n$ Haar or SRHT matrix. Our first results concern Haar projections (Section 3). By leveraging results about their limiting spectral distributions, and after some calculations with Stieljes transforms (defined below) we provide the following new trace formula (see Lemma 3.2):

$$\theta_{2,h} := \lim_{n \to \infty} \frac{1}{d} \operatorname{tr} \mathbb{E} \left[ (U^\top S^\top S U)^{-2} \right] = \frac{(1-\gamma)(\gamma^2 + \xi - 2\gamma\xi)}{(\xi - \gamma)^3} \, .$$

As an application, we characterize explicitly the optimal step sizes $\mu_t$ and momentum parameters $\beta_t$ of the IHS with Haar embeddings (Theorem 3.1). We emphasize that the optimal parameters have asymptotically closed form for any data matrix $A$, unlike for certain other propular methods such as gradient descent, which can be useful in practice. With these optimal parameters, we find that at *any* time step $t \geqslant 1$ (Theorem 3.1),

$$\lim_{n \to \infty} \frac{\mathbb{E}\|\Delta_t\|^2}{\|\Delta_0\|^2} = \rho_h^t \, , \tag{3}$$

where the convergence rate $\rho_h$ is given by $\rho_h := \rho_g \cdot \frac{\xi(1-\xi)}{\gamma^2 + \xi - 2\xi\gamma}$, and always satisfies $\rho_h < \rho_g$. By comparing with the prior work [13], this implies that Haar embeddings have uniformly better performance than Gaussian ones. Further, as an immediate consequence of Theorem 2 in [13], we obtain that the optimal momentum parameters $\beta_t$ are equal to 0, that is, Heavy-ball momentum does not accelerate the algorithm with refreshed Haar embeddings (Theorem 3.1 and following discussion). Thus, we are able to characterize explicitly the optimal version of the IHS with Haar embeddings.

Our next results concern SRHT sketches (Section 4). We prove that under the additional mild assumption on the initial error $\Delta_0$ that $\mathbb{E}[\Delta_0 \Delta_0^\top] = d^{-1} I_d$, the IHS with SRHT embeddings also has rate of convergence $\rho_h$ (Theorem 4.1). This relies on novel formulas for the first two inverse moments of SRHT sketches (Lemma 4.3). Consequently, *SRHT matrices uniformly outperform Gaussian embeddings*. Then, we confirm numerically the above theoretical statements (Section 6).

We finally analyze the computational complexity of our method, in comparison to some standard randomized pre-conditioned solvers [26] for dense, ill-conditioned least-squares. We show that in our infinite-dimensional regime, we improve by a factor $\log d$ (Section 5).

Importantly, we specifically focus on the IHS with refreshed i.i.d. embeddings. An immediate variant of the IHS uses the same update (2), but with a fixed embedding $S$ drawn only once at the first iteration, which is appealing in practice. In a concurrent paper [15] more recent to the initial version of the present work, it has been shown that, in the same asymptotic regime, the IHS with a fixed SRHT embedding achieves a better convergence rate. Thus, we emphasize that our core contributions are to develop novel techniques and results for analyzing the IHS with the SRHT, as this may be useful for future developments and extensions of this algorithm in different contexts (e.g., constrained least-squares, convex optimization).

Although we characterize the optimal step sizes and momentum parameters for the IHS with Haar embeddings, we only characterize the optimal step size in the absence of momentum for the IHS with the SRHT. It is thus left as an open question to know whether momentum can accelerate further our method.

## 2 Technical Background

We introduce a few needed definitions, and we refer the reader to [5, 3, 22, 38] for an extensive introduction to random matrix theory. Let $\{M_n\}_n$ be a sequence of Hermitian random matrices, where each $M_n$ has size $n \times n$. For a fixed $n$, the empirical spectral distribution (e.s.d.) of $M_n$ is the (cumulative) distribution function of its eigenvalues $\lambda_1, \ldots, \lambda_n$, i.e., $F_{M_n}(x) := \frac{1}{n} \sum_{j=1}^n \mathbf{1}\{\lambda_j \leqslant x\}$ for $x \in \mathbb{R}$, which has density $f_{M_n}(x) = \frac{1}{n} \sum_{j=1}^n \delta_{\lambda_j}(x)$ with $\delta_\lambda$ the Dirac measure at $\lambda$. Due to the randomness of the eigenvalues, $F_{M_n}$ is random. The relevant aspect of some classes of large $n \times n$ symmetric random matrices $M_n$ is that, almost surely, the e.s.d. $F_{M_n}$ converges weakly towards

a non-random distribution $F$, as $n \to \infty$. This function $F$, if it exists, will be called the *limiting spectral distribution* (l.s.d.) of $M_n$.

A powerful tool in the analysis of random matrices is the Stieltjes transform. For $\mu$ a probability measure supported on $[0, +\infty)$, its Stieltjes transform is defined over the complex space complementary to the support of $\mu$ as

$$m_\mu(z) := \int \frac{1}{x - z} \, \mathrm{d}\mu(x) . \tag{4}$$

It holds in particular that $m_\mu$ is analytic over $\mathbb{C} \setminus \mathbb{R}_+$, $m_\mu(z) \in \mathbb{C}^+$ for $z \in \mathbb{C}^+$, $m_\mu(z) \in \mathbb{C}^-$ for $z \in \mathbb{C}^-$ and $\mu_\mu(z) > 0$ for $z < 0$, where $\mathbb{R}_+$ is the set of positive reals and $\mathbb{C}^+$ is the set of complex numbers with positive imaginary part. Another useful transform for studying the product of random matrices is the $S$-transform, denoted $S_\mu$. This is defined as the solution of the following equation, which is unique under certain conditions (see [33]),

$$m_\mu\left(\frac{z+1}{zS_\mu(z)}\right) + zS_\mu(z) = 0. \tag{5}$$

We introduce a few additional concepts from free probability that will be used in the proofs. We refer the reader to [33, 12, 21, 3] for an extensive introduction to this field. Consider the algebra $\mathcal{A}_n$ of $n \times n$ random matrices. For $X_n \in \mathcal{A}_n$, we define the linear functional $\tau_n(X_n) := \frac{1}{n}\mathbb{E}[\text{trace } X_n]$. Then, we say that a family $\{X_{n,1}, \ldots, X_{n,I}\}$ of random matrices in $\mathcal{A}_n$ is *asymptotically free* if for every $i \in \{1, \ldots, I\}$, $X_{n,i}$ has a limiting spectral distribution, and if $\tau\left(\prod_{j=1}^m P_j\left(X_{n,i_j} - \tau\left(P_j(X_{n,i_j})\right)\right)\right) \to 0$ almost surely for any positive integer $m$, any polynomials $P_1, \ldots, P_m$ and any indices $i_1, \ldots, i_m \in \{1, \ldots, I\}$ with $i_1 \neq i_2, \ldots, i_{m-1} \neq i_m \neq i_1$. In particular, this definition implies that for two sequences of asymptotically free random matrices $X_n, Y_n$, we have the *trace decoupling* relation

$$\frac{1}{n}\mathbb{E}[\text{trace } X_n Y_n] - \frac{1}{n}\mathbb{E}[\text{trace } X_n]\frac{1}{n}\mathbb{E}[\text{trace } Y_n] \to 0 . \tag{6}$$

Essential to our analysis is the following result. If two $n \times n$ random matrices $A_n$ and $B_n$ are asymptotically free and have respective l.s.d. $\mu_A$ and $\mu_B$ with respective $S$-transforms $S_A$ and $S_B$, then the matrix product $A_n B_n$ has l.s.d. $\mu_{AB}$ whose $S$-transform is $S_{AB}(z) = S_A(z)S_B(z)$. The distribution $\mu_{AB}$ is called the free multiplicative convolution of $\mu_A$ and $\mu_B$, and we denote $\mu_{AB} = \mu_A \boxtimes \mu_B$.

We will also make use of an alternative form of the Stieltjes transform: the $\eta$-transform is defined for $z \in \mathbb{C} \setminus \mathbb{R}^-$ as

$$\eta_\mu(z) := \int \frac{1}{1 + zx} \, \mathrm{d}\mu(x) = \frac{1}{z}m_\mu\left(-\frac{1}{z}\right) . \tag{7}$$

There are standard examples of classes of random matrices and their limiting spectral behavior. We recall a classical result [19]. If $S$ is an $m \times d$ matrix with identically and independently distributed entries $\mathcal{N}(0, 1/m)$, then, as $m, d \to \infty$ with $m/d \to \rho \in (0, 1)$, the Marchenko-Pastur theorem (see [19, 5]) states that the matrix $S^\top S$ has l.s.d. $F_\rho$, whose Stieltjes transform is the unique solution of a certain fixed point equation, and whose density is explicitly given by

$$\mu_\rho(x) = \frac{\sqrt{(b - x)_+(x - a)_+}}{2\pi\rho x} , \tag{8}$$

where $y_+ = \max\{0, y\}$, $a = (1 - \sqrt{\rho})^2$ and $b = (1 + \sqrt{\rho})^2$. In our analysis of Haar and SRHT matrices, we will encounter similar fixed-point equations satisfied by the Stieltjes (or $\eta$-) transform of their l.s.d.

## 3 Sketching with Haar matrices

Sketching matrices with i.i.d. entries are not ideal for sketching. Intuitively, i.i.d. projections distort the geometry of Euclidean space due to their non-orthogonality. In this section, we consider the IHS with refreshed Haar matrices $\{S_t\}$. The following result says that orthogonal projection has better performance than Gaussian projection.

**Theorem 3.1** (Optimal IHS with Haar sketches). *With refreshed Haar matrices $\{S_t\}$, step sizes $\mu_t = \theta_{1,h}/\theta_{2,h}$ (where $\theta_{i,h}$ are defined in Lemma 3.2) and momentum parameters $\beta_t = 0$, the sequence of error vectors $\{\Delta_t\}$ satisfies*

$$\rho_h := \left( \lim_{n \to \infty} \frac{\mathbb{E}\|\Delta_t\|^2}{\|\Delta_0\|^2} \right)^{1/t} = \rho_g \cdot \frac{\xi(1-\xi)}{\gamma^2 + \xi - 2\xi\gamma} . \tag{9}$$

*Further, for any sequence of step sizes $\{\mu_t\}$ and momentum parameters $\{\beta_t\}$, we have that, for the resulting sequence of error vectors $\{\Delta_t\}$,*

$$\rho_h \leqslant \liminf_{t \to \infty} \left( \lim_{n \to \infty} \frac{\mathbb{E}\|\Delta_t\|^2}{\|\Delta_0\|^2} \right)^{1/t} , \tag{10}$$

*that is, $\rho_h$ is the optimal rate one may achieve using Haar embeddings.*

The proof of Theorem 3.1, whose details are deferred to Appendix A.2, is decomposed into two steps. First, we relate the asymptotic convergence rate $\rho_h$ to the first and second moments of the inverse l.s.d. of the sketched matrix $SU$, and we adapt to the asymptotic setting the proof of Theorem 1 in [13]. Then, and this is our key technical contribution, we provide an explicit formula of this second moment, as given in the following technical lemma.

**Lemma 3.2** (First two inverse moments of Haar sketches). *Suppose that $S$ is an $m \times n$ Haar matrix, and let $U$ be an $n \times d$ deterministic matrix with orthonormal columns. It holds that*

$$\theta_{1,h} := \lim_{n \to \infty} \frac{1}{d} \text{ trace } \mathbb{E}\left[ (U^\top S^\top S U)^{-1} \right] = \frac{1-\gamma}{\xi - \gamma}$$

$$\theta_{2,h} := \lim_{n \to \infty} \frac{1}{d} \text{ trace } \mathbb{E}\left[ (U^\top S^\top S U)^{-2} \right] = \frac{(1-\gamma)(\gamma^2 + \xi - 2\gamma\xi)}{(\xi - \gamma)^3} . \tag{11}$$

The formula of the second moment, to the best of our knowledge, is derived explicitly for the first time. We provide a proof sketch here. Note that $\theta_{i,h}$ ($i = 1, 2$) is the average of the eigenvalues of $U^\top S^\top S U$ to the power of $-i$. Denoting $F_h$ the limiting distribution of the eigenvalues of $U^\top S^\top S U$, we have $\theta_{i,h} = \int x^{-i} dF_h(x)$. This matrix has a specific structure whose l.s.d. has been studied in the random matrix literature. Specifically, given some diagonal non-negative matrices $D, T$ and a squared Haar matrix $W$, Theorem 4.11 of [7] characterizes the l.s.d. of matrices of the form $D^{\frac{1}{2}} W T W^\top D^{\frac{1}{2}}$ through a system of functions involving its $\eta$-transform and the l.s.d. of $D, T$. Our setting is more intricate, as $S, U$ are both partial orthogonal matrices, and we need to use an orthogonal complement trick. After getting the $\eta$-transform and thus the Stieltjes transform $m(z) = \int \frac{1}{x-z} dF_h(x)$, we can calculate $\theta_{1,h}, \theta_{2,h}$ by evaluating the first and second derivatives of $m(z)$ at 0. Fortunately in our case, the Stieltjes transform has a closed form, though the calculation is cumbersome. We defer the detailed proof to Appendix A.1.

One might wonder how the l.s.d. of Haar matrices and that of Gaussian embeddings – the Marchenko-Pastur law $\mu_{\rho_g}$ – differ. Consider the re-scaled matrix $\frac{n}{m} S_{1,n}^\top S_{1,n}$, whose expectation is equal to the identity. Crucially, the l.s.d. $\mu_{\rho_g}$ does not depend on the sample size $n$ but only on the limit ratio between $d$ and $m$, whereas the distribution $F_h$ involves the ratios $\gamma$ and $\xi$. Numerically, we observe in Figure 1 that, for fixed $\gamma = 0.2$, as $\xi$ increases, the empirical Haar density departs from the Marchenko-Pastur density $\mu_{\rho_g}$, and concentrates more and more relatively to $\mu_{\rho_g}$. Importantly, we see that the support of $F_h$ is included within the support of $\mu_{\rho_g}$, and thus, more concentrated around 1. According to Theorem 3.1 orthogonal projections are uniformly better than Gaussian i.i.d. projections. Indeed, the ratio between the convergence rates $\rho_h$ and $\rho_g$ is equal to $\xi(1-\xi)/(\gamma^2 + \xi - 2\gamma\xi)$, and is *always strictly smaller* than 1. To see this, note that $\xi(1-\xi)/(\gamma^2 + \xi - 2\gamma\xi) < 1$ if and only if $\xi(1-\xi) < \gamma^2 + \xi - 2\gamma\xi$, and after simplification, we obtain the condition $(\xi - \gamma)^2 > 0$. In the small sketch size regime $d \leqslant m \ll n$, we have $\rho_h/\rho_g \approx 1$. As the sketch size $m$ increases relatively to $n$, the convergence rates' ratio scales as $\rho_h/\rho_g \approx (1 - \xi)$, and one can improve on the number of iterations – and thus, data passes – with Haar embeddings by making $1 - \xi$ bounded away from 1. Further, observe that if we do not reduce the size of the original matrix, so that $m = n$ and $\xi = 1$, then the algorithm converges in one iteration. This means that we do not lose any information in the linear model. In contrast, Gaussian projections introduce more distortions than rotation, even though the rows of a Gaussian matrix are almost orthogonal to each other in the high-dimensional setting. The reason is that the eigenvalues are not close to unity.

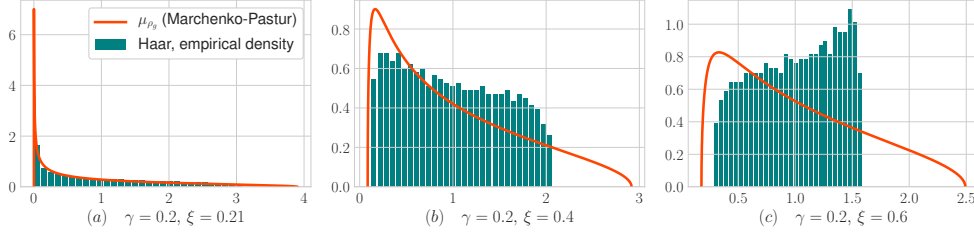

Figure 1: Empirical density of the matrix $\frac{n}{m}U^\top S^\top S U$ for $S$ an $m \times n$ Haar matrix, versus Marchenko-Pastur density with shape parameter $d/m$. We use $n = 4096$, $d = 820$ and $m \in \{860, 1640, 2450\}$, so that $\gamma \approx 0.2$ and $\xi \in \{0.2, 0.4, 0.6\}$.

Interestingly, momentum does not accelerate the refreshed sketch with Haar embeddings. Leveraging past information through the Heavy-ball update (2) does not provide any benefit, possibly due to the independence between the sketching matrices $\{S_t\}$. Our proof of this fact is actually an immediate consequence of Theorem 2 in [13]. On the other hand, it remains an open question whether there exists a first-order method which uses past iterates along with refreshed matrices, and provide acceleration over gradient descent updates.

We also emphasize that *the optimal parameters have asymptotically closed forms, for any data matrix $A$!* This is quite unexpected and can be useful in practice. The reason is that random projections introduce a great deal of regularity, leading to a "universal" behavior of certain quantities, including those we need. For methods such as gradient descent with momentum, the optimal parameters (e.g, stepsize, momentum), can depend on quantities that can be nontrivial to estimate (e.g, the Lipschitz constant), and require extra computational work.

However, the time complexity of generating an $m \times n$ Haar matrix using the Gram-Schmidt procedure is $O(nm^2)$, which is, for instance, larger than the classical cost $\mathcal{O}(nd^2)$ for solving the least-squares problem (1), and we now turn to the analysis of another orthogonal matrix, the SRHT, which contains less randomness, but is more structured and faster to generate.

## 4 Sketching with SRHT matrices

We have seen in the previous section that Haar random projections have a better performance than Gaussian i.i.d. random projections. However, they are still slow to generate and apply. Can we get the same good statistical performance as Haar projections with faster methods? Here we consider the SRHT. This is faster as it relies on the well-structured Walsh-Hadamard transform, which is defined as follows. For an integer $n = 2^p$ with $p \geqslant 1$, the Walsh-Hadamard transform is defined recursively as $H_n = \begin{bmatrix} H_{n/2} & H_{n/2} \\ H_{n/2} & -H_{n/2} \end{bmatrix}$ with $H_1 = 1$. We consider a version of the SRHT which is slightly different from the classical SRHT [1]. Our transform $A \mapsto SA$ first *randomly permutes* the rows of $A$, before applying the classical transform. This has negligible cost $\mathcal{O}(n)$ compared to the cost $\mathcal{O}(nd \log m)$ of the matrix multiplication $A \mapsto SA$, and *breaks the non-uniformity* in the data. That is, we define the $n \times n$ subsampled randomized Hadamard matrix as $S = BH_nDP/\sqrt{n}$, where $B$ is an $n \times n$ diagonal sampling matrix of i.i.d. Bernoulli random variables with success probability $m/n$, $H_n$ is the $n \times n$ Walsh-Hadamard matrix, $D$ is an $n \times n$ diagonal matrix of i.i.d. sign random variables, equal to $\pm 1$ with equal probability, and $P \in \mathbb{R}^{n \times n}$ is a uniformly distributed permutation matrix. At the last step, we discard the zero rows of $S$, so that it becomes an $\widetilde{m} \times n$ orthogonal matrix with $\widetilde{m} \sim \text{Binomial}(m/n, n)$, and the ratio $\widetilde{m}/n$ concentrates fast around $\xi$ as $n \to \infty$. Although the dimension $\widetilde{m}$ is random, we refer to $S$ as an $m \times n$ SRHT matrix.

The following theorem characterizes the exact convergence rate of the IHS with refreshed SRHT embeddings.

**Theorem 4.1** (IHS with SRHT sketches). *Suppose that the initial point $x_0$ is random and that the error vector $\Delta_0$ satisfies the condition $\mathbb{E}\left[\Delta_0\Delta_0^\top\right] = d^{-1}I_d$. Then, with refreshed SRHT matrices $\{S_t\}$, step sizes $\mu_t = \theta_1^h/\theta_2^h$ and momentum parameters $\beta_t = 0$, the sequence of error vectors $\{\Delta_t\}$*

*satisfies*

$$\rho_s := \left( \lim_{n \to \infty} \frac{\mathbb{E}\|\Delta_t\|^2}{\mathbb{E}\|\Delta_0\|^2} \right)^{1/t} = \rho_g \cdot \frac{\xi(1-\xi)}{\gamma^2 + \xi - 2\xi\gamma} = \rho_h \,. \tag{12}$$

Here we impose an additional mild assumption on the initialization of the least-squares problem (1). We note that the initialization condition $\mathbb{E}\left[\Delta_0 \Delta_0^\top\right] = d^{-1} I_d$ can be achieved by picking $x_0$ uniformly on the unit $d$-sphere $\mathbb{S}^{d-1}$, followed by a uniformly random signed permutation and scaling to the columns of $A$. The key challenge to avoid this is that we need to evaluate $\mathbb{E}\left[\|\Delta_t\|^2\right] = \operatorname{trace} \mathbb{E}\left[Q_0 \ldots Q_{t-1} Q_{t-1} \ldots Q_0 \Delta_0 \Delta_0^\top\right]$, where $Q_t = I_d - \mu_t \, (U^\top S_t^\top S_t U)^{-1}$ and $U$ are the left singular vectors of $A$. Understanding this for general $\Delta_0$ requires properties that are not currently known in random matrix theory (see Appendix A.4 and Remark A.1 for more details). Further we can only analyze the case $\beta_t = 0$, and we do not have a proof for optimality, but *we conjecture that it is true* based on numerical simulations.

We also present an upper-bound on the error, which holds for any deterministic or random initialization $x_0$ and exhibits an identical convergence rate. This is weaker by a factor of $d$, but this is negligible for large $t$.

**Theorem 4.2.** *For any initialization $x_0$, with refreshed SRHT matrices $\{S_t\}$, step sizes $\mu_t = \theta_1^h / \theta_2^h$ and momentum parameters $\beta_t = 0$, the sequence of error vectors $\{\Delta_t\}$ satisfies*

$$\limsup_{n \to \infty} \left( \frac{\mathbb{E}\|\Delta_t\|^2}{d \cdot \mathbb{E}\|\Delta_0\|^2} \right)^{1/t} \le \rho_h \,. \tag{13}$$

The proofs of Theorem 4.1 and 4.2 are deferred to Appendix A.4. While providing significant computational benefits for forming the sketch $SA$, SRHT embeddings are still able to match the convergence rate of orthogonal projections, and thus, also improves on Gaussian sketches. This result follows from the observation that, althouth SRHT has much less randomness than Haar projection, their first two inverse moments behave the same asymptotically. This is formally stated in the following lemma.

**Lemma 4.3** (First two inverse moments of SRHT sketches). *Let $S$ be an $m \times n$ SRHT matrix, $S_h$ be an $m \times n$ Haar matrix, and $U$ an $n \times d$ deterministic matrix with orthonormal columns. Then, the matrices $U^\top S^\top S U$ and $U^\top S_h^\top S_h U$ have the same limiting spectral distribution. Consequently, with $\theta_{1,h}, \theta_{2,h}$ as defined in Lemma 3.2, it holds that*

$$\lim_{n \to \infty} \frac{1}{d} \operatorname{trace} \mathbb{E}\left[(U^\top S^\top S U)^{-1}\right] = \theta_{1,h} \,, \tag{14}$$

$$\lim_{n \to \infty} \frac{1}{d} \operatorname{trace} \mathbb{E}\left[(U^\top S^\top S U)^{-2}\right] = \theta_{2,h} \,. \tag{15}$$

The proof is based on recent results about *asymptotically liberating sequences* from the free probability literature [2], which proves the asymptotic freeness for Hadamard matrices. This technique is also used in [8] to study SRHT. Specifically, they defined the bi-signed-permutation Hadamard matrix $W = P^\top D H D P$, where $H$ is a Hadamard matrix, $D$ is a sign-flipping diagonal matrix, and $P$ is a permutation. Corollary 3.5, 3.7 of [2] showed that the Bernoulli-sampling diagonal matrix $B$ and $W U U^\top W$ are asymptotically free in the non-commutative probability space of random matrices. Another observation is that, by changing the definition of $S$ to $S = B P^\top D H D P = B W$, the l.s.d. of $U^\top S^\top S U$ remain the same as when $S = B H D P$. The asymptotic freeness shows that the l.s.d. of $U^\top S^\top S U$ for $S$ an SRHT is the same as when $S$ is a Haar matrix. So we get the same results as in Lemma 3.2. The detailed proof is defered to Appendix A.3.

In Figure 2, we verify that the empirical densities with Haar and SRHT matrices are indeed very close.

## 5   Complexity Analysis

Let us now turn to a complexity analysis of the IHS with SRHT embeddings, and compare it, in an asymptotic sense, to the complexity of the standard pre-conditioned conjugate gradient method [26].

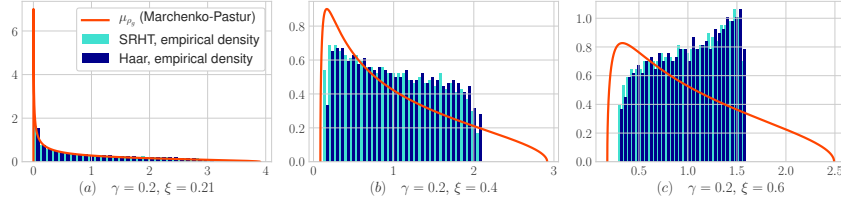

Figure 2: Empirical densities of the matrices $\frac{n}{m} U^\top S^\top S U$ for $S$ an $m \times n$ Haar matrix and SRHT matrix, versus Marchenko-Pastur density with shape parameter $d/m$. We use $n = 4096$, $d = 820$ and $m \in \{860, 1640, 2450\}$, so that $\gamma \approx 0.2$ and $\xi \in \{0.21, 0.4, 0.6\}$.

The latter uses a sketch $SA$ to compute a pre-conditioning matrix $P$, such that $AP^{-1}$ has a small condition number, and then it solves the least-squares problem $\min_y \|AP^{-1}y - b\|^2$, using the conjugate-gradient method. As for the IHS, it can be decomposed into three parts: sketching, factoring (computing $P$ and $AP^{-1}$ versus computing $H_t$), and iterating. The pre-conditioned conjugate gradient prescribes the sketch size $m \approx d \log d$ to guarantee convergence with high-probability. This lower bound is based on the finite-sample bounds on the extremal eigenvalues of the matrix $U^\top S^\top S U$ derived by [30]. Then, given $\varepsilon > 0$ and with $m \approx d \log d$, the resulting complexity to achieve $\|\Delta_t\|^2 \leqslant \varepsilon$ scales as $\mathcal{C}_c \asymp nd \log d + d^3 \log d + nd \log(1/\varepsilon)$, where $nd \log d$ is the cost of forming $SA$, the term $d^3 \log d$ is the factoring cost, and $nd \log(1/\varepsilon)$ is the per-iteration cost times the number of iterations. In contrast, we obtain that the IHS with the SRHT can use $m \approx d$, with resulting complexity $\mathcal{C}_n \asymp (nd \log d + d^3 + nd) \log(1/\varepsilon)$. Note that the number of iterations multiplies the sum of the sketching, factoring and per-iteration costs, and this is due to refreshing the sketches. Then, treating the term $\log(1/\varepsilon)$ as a constant independent of the dimensions, we find that, as $n, d, m$ grow to infinity, we have that $C_n/C_c \asymp 1/\log d$.

## 6 Numerical Simulations

### 6.1 Comparison of the different variants of the iterative Hessian sketch

We evaluate the performance of the IHS with refreshed Haar/SRHT sketches against refreshed Gaussian sketches.

First, we generate a synthetic data matrix $A \in \mathbb{R}^{n \times d}$ with exponential spectral decay (its $j$-th singular value of $A$ is $\sigma_j = 0.98^j$) and where $n = 8192$ and $d = 800$. We consider the sketch sizes $m \in \{980, 2450, 4100\}$. For the SRHT, we use the step size $\mu_t = \theta_{1,h}/\theta_{2,h}$ prescribed in Theorem 4.1, where we replace $\xi$ and $\gamma$ by their finite sample approximations $\xi \approx \frac{m}{n}$ and $\gamma \approx \frac{d}{n}$. For refreshed Gaussian embeddings, we use the optimal parameters $\mu_t$ and $\beta_t$ derived in [13]. Results are reported in Figure 3. As $m$ increases, Haar/SRHT embeddings are increasingly better compared to Gaussian projections. Further, the empirical curves match closely our theoretical predictions: the algorithmic parameters derived from our asymptotic analysis are useful in practice when they are replaced by their finite-sample approximations. Second, we carry out a similar experiment with

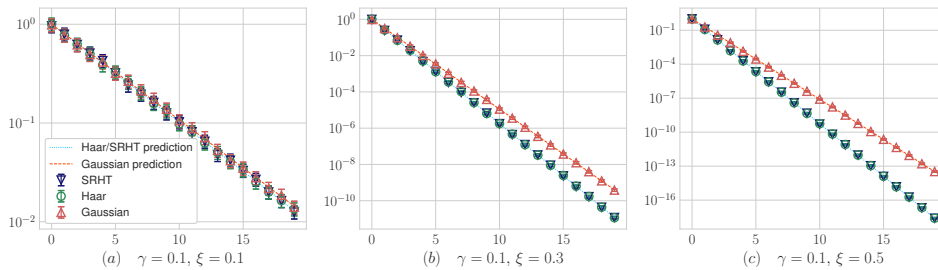

Figure 3: Synthetic dataset: Error $\|\Delta_t\|^2/\|\Delta_0\|^2$ versus number of iterations for the iterative Hessian sketch: (a) $m = 980$, (b) $m = 2450$ and (c) $m = 4100$. We average over 50 independent trials and empirical standard deviations are shown in the form of error bars.

the CIFAR10 dataset, for which we consider one-vs-all classification. Here, we have $n = 60000$, $d = 3072$ and we use the sketch sizes $m \in \{6000, 18000, 30000\}$. Results are reported in 4, and we observe similar quantitative results as for the aforementioned synthetic dataset.

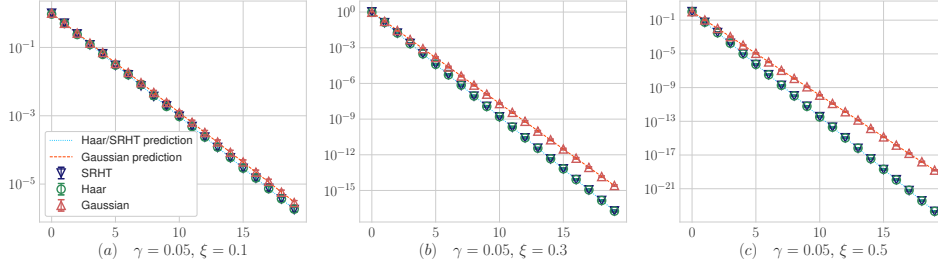

Figure 4: CIFAR10 dataset: Error $\|\Delta_t\|^2/\|\Delta_0\|^2$ versus number of iterations for the iterative Hessian sketch: (a) $m = 6000$, (b) $m = 18000$ and (c) $m = 30000$. We average over 50 independent trials and empirical standard deviations are shown in the form of error bars.

## 6.2 Comparison of the iterative Hessian sketch to standard iterative solvers

We compare the IHS with the SRHT against the conjugate gradient (CG) method and its preconditioned (pCG) version [26]. We also consider a variant of the IHS, for which we do not refresh the embedding at every iteration. We generate a synthetic data matrix $A \in \mathbb{R}^{n \times d}$ with exponential spectral decay ($\sigma_j = 0.98^j$), $n = 4096$ and $d = 200$. We consider the sketch sizes $m \in \{1000, 1500, 2000\}$. We observe that the IHS which refreshes embeddings at every iteration has the best convergence rate. More generally, the higher this update frequency, the better the performance. In comparison, CG has the worst convergence rate, which is expected since the data matrix is ill-conditioned, and pCG performs slightly worse than the IHS with update frequency equal to $1$.

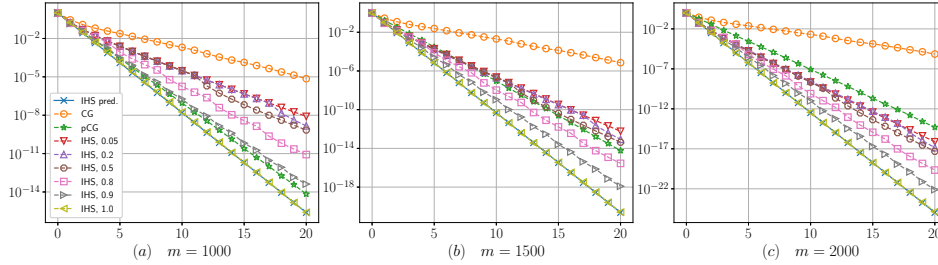

Figure 5: Error $\|\Delta_t\|^2/\|\Delta_0\|^2$ versus number of iterations for the iterative Hessian sketch with the SRHT and different sketch sizes. We average over 50 independent trials. For instance, 'IHS, 0.2' refers to the IHS with update frequency equal to $0.2$. For clarity, we do not show error bars for the mean empirical standard deviation which are barely visible.

## Broader Impact

We believe that the proposed method in this work can have positive societal impacts. Our algorithm can be applied in massive scale distributed learning and optimization problems encountered in real-life problems. The computational effort can be significantly lowered as a result of dimension reduction. Consequently energy costs for optimization can be significantly reduced.

## Acknowledgments and Disclosure of Funding

This work was partially supported by the National Science Foundation under grants IIS-1838179 and ECCS-2037304, Facebook Research, Adobe Research and Stanford SystemX Alliance.

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
