[Supplementary Material]

# Supplement to "Optimal Iterative Sketching Methods with the Subsampled Randomized Hadamard Transform"

## Contents

## A  Proofs of main theorems

### A.1  Calculations of $\theta_{1,h}$ and $\theta_{2,h}$ for Haar sketch

We first prove some lemmas and provide the proof of 3.2 in Section A.1.1.

This lemma characterizes the Stieltjes transform of the l.s.d. of $S_n U_n$.

**Lemma A.1** (Stieltjes transform of l.s.d. of $S_n U_n$). *We set $S_{1,n} = S_n U_n$. Then the matrix $S_{1,n}^\top S_{1,n}$ admits a l.s.d. whose Stieltjes transform $m_h$ is given by*

$$m_h(z) = \frac{z(2\gamma-1) + \xi - \gamma - \sqrt{(\gamma+\xi-2+z)^2 + 4(z-1)(1-\gamma)(1-\xi)}}{2\gamma z(1-z)}, \tag{1}$$

*for any $z \in \mathbb{C} \setminus \mathbb{R}_+$.*

*Proof.* First, observe that since both $S_n$ and $U_n$ are rectangular orthogonal matrices, we can embed them into full orthogonal matrices as $\mathbb{S}_n = \begin{pmatrix} S_n \\ S_n^\perp \end{pmatrix}$ and $\mathbb{U}_n = \begin{pmatrix} U_n & U_n^\perp \end{pmatrix}$. Then, we can write

$$S_{1,n} = \begin{pmatrix} I_m & 0 \end{pmatrix} \mathbb{S}_n \mathbb{U}_n \begin{pmatrix} I_d \\ 0 \end{pmatrix}. \tag{2}$$

Let $\mathbb{W}_n = \mathbb{S}_n \mathbb{U}_n$, which is an $n \times n$ Haar matrix due to the orthogonal invariance of the Haar distribution. Then, we define

$$C_n := \begin{pmatrix} S_{1,n} S_{1,n}^\top & 0 \\ 0 & 0 \end{pmatrix} = \begin{pmatrix} I_m & 0 \\ 0 & 0 \end{pmatrix} \mathbb{W}_n \begin{pmatrix} I_d & 0 \\ 0 & 0 \end{pmatrix} \mathbb{W}_n^\top \begin{pmatrix} I_m & 0 \\ 0 & 0 \end{pmatrix}. \tag{3}$$

The matrix $C_n$ is related to our matrix of interest $S_{1,n}^\top S_{1,n}$, as they have exactly the same non-zero eigenvalues. Thus, as a first step to establish Lemma A.1, we characterize the l.s.d. of $C_n$.

The matrix $C_n$ admits a l.s.d. $F_C$, whose Stieltjes transform $m_C$ is given by

$$m_C(z) = \frac{z + \gamma + \xi - 2 - \sqrt{(\gamma + \xi - 2 + z)^2 + 4(z-1)(1-\gamma)(1-\xi)}}{2z(1-z)}, \tag{4}$$

for any $z \in \mathbb{C} \setminus \mathbb{R}_+$. The above expression (3) of the matrix $C_n$ has the required form to apply Theorem 4.11 by [2], and hence characterize the e.s.d. of $C_n$ through its $\eta$-transform which has to satisfy a fixed-point equation. We defer details of the proof to Section B.2. Now, we use the fact that the matrices $S_{1,n}^\top S_{1,n}$ and $C_n$ have the same non-zero eigenvalues. Almost surely, there are exactly $d$ of them, which we denote $\lambda_1, \ldots, \lambda_d$. Then, the e.s.d. $F_{C_n}$ of $C_n$ can be decomposed as

$$F_{C_n}(x) = \left(1 - \frac{d}{n}\right) \mathbf{1}_{\{x \geqslant 0\}} + \frac{1}{n} \sum_{i=1}^{d} \mathbf{1}_{\{x \geqslant \lambda_i\}} = \left(1 - \frac{d}{n}\right) \mathbf{1}_{\{x \geqslant 0\}} + \frac{d}{n} F_{h,n}(x), \tag{5}$$

where $F_{h,n}$ is the e.s.d. of $S_{1,n}^\top S_{1,n}$. Taking the limit $n \to \infty$, we find that $F_{1,n}$ converges weakly almost surely to

$$F_h(x) = \frac{1}{\gamma} \left(F_C(x) - (1-\gamma)\mathbf{1}_{\{x \geqslant 0\}}\right). \tag{6}$$

By definition of $m_h$ and using (6), it follows that for $z \in \mathbb{C} \setminus \mathbb{R}_+$

$$m_h(z) = \int \frac{1}{x-z} \, \mathrm{d}F_h(x) = \frac{1}{\gamma} \int \frac{1}{x-z} \, \mathrm{d}F_C(x) - \frac{1-\gamma}{\gamma} \int \frac{1}{x-z} \delta_0(x)\mathrm{d}x \tag{7}$$

$$= \frac{1}{\gamma} m_C(z) + \frac{1-\gamma}{\gamma z}. \tag{8}$$

Plugging-in the expression of $m_C$, we obtain the claimed formula (1) for $m_h$.

$\square$

We will need the following result regarding the support of $F_h$, which is proved in Appendix B.1.

**Lemma A.2.** *The support of $F_h$ satisfies*

$$\inf \mathrm{supp}(F_h) \geqslant \frac{(1 - \sqrt{\rho_g})^2}{\left(1 + \frac{1}{\sqrt{\xi}}\right)^2}. \tag{9}$$

Thus, the support of $F_h$ is bounded away from 0, so is the intersection of the support of $F_C$ and $\mathbb{R}^*$. Further, the distribution $F_C$ has a point mass at 0 equal to $1 - \gamma$. We now turn to the trace calculations in Lemma 3.2.

### A.1.1 Proof of Lemma 3.2

1. **Computing $\theta_{1,h}$**

   Using the facts that $F_C$ has support within $[0, +\infty)$ and a point mass equal to $(1-\gamma)$ at 0, its $\eta$-transform $\eta_C$ is well-defined on $\{z \in \mathbb{R} \mid z > 0\}$, and, for $z > 0$, it can be decomposed as

   $$\eta_C(z) = 1 - \gamma + \int_{x \neq 0} \frac{1}{1+zx} \mathrm{d}F_C(x). \tag{10}$$

   The function $\frac{1}{x}$ is integrable on the set $\{x > 0\}$ with respect to $F_C$, since the support of $F_C$ on $\mathbb{R}^*$ is bounded away from 0. Since $\left|\frac{z}{1+xz}\right| < \frac{1}{x}$ when $z > 0, x > 0$, it follows by the dominated convergence theorem that

   $$\lim_{z \to \infty} \int_{x \neq 0} \frac{z}{1+xz} \mathrm{d}F_C(x) = \int_{x \neq 0} \lim_{z \to \infty} \frac{z}{1+xz} \mathrm{d}F_C(x) = \int_{x \neq 0} \frac{1}{x} \mathrm{d}F_C(x). \tag{11}$$

Using (10), it follows that

$$\lim_{z \to \infty} z \left( \eta_C(z) - (1 - \gamma) \right) = \int_{x \neq 0} \frac{1}{x} \, \mathrm{d}F_C(x), \tag{12}$$

On the other hand, we have that

$$\lim_{z \to \infty} \eta_C(z) = (1 - \gamma) + \lim_{z \to \infty} \int_{x \neq 0} \frac{1}{1 + zx} \, \mathrm{d}F_C(t) \tag{13}$$

$$= (1 - \gamma) + \int_{x \neq 0} \lim_{z \to \infty} \frac{1}{1 + zx} \, \mathrm{d}F_C(x) \tag{14}$$

$$= 1 - \gamma. \tag{15}$$

where the second equality is again justified by the dominated convergence theorem. Subtracting $1 - \gamma$ from both sides of (33), multiplying by $z \left( 1 + \frac{\xi - 1}{\eta_C(z)} \right)$ and letting $z \to \infty$, we obtain

$$\lim_{z \to \infty} z \left( 1 + \frac{\xi - 1}{\eta_C(z)} \right) \left( \eta_C(z) - (1 - \gamma) \right) = \lim_{z \to \infty} z \left( 1 + \frac{\xi - 1}{\eta_C(z)} \right) \left( \frac{\gamma}{1 + z(1 + \frac{\xi - 1}{\eta_C(z)})} \right).$$

Note that the right-hand side of the above equation is equal to $\gamma$, and the left-hand side satisfies

$$\lim_{z \to \infty} z \left( 1 + \frac{\xi - 1}{\eta_C(z)} \right) \left( \eta_C(z) - (1 - \gamma) \right) = \lim_{z \to \infty} z \left( \eta_C(z) - (1 - \gamma) \right) \left( 1 + \frac{\xi - 1}{1 - \gamma} \right)$$

$$= \frac{\xi - \gamma}{1 - \gamma} \cdot \int_{x \neq 0} \frac{1}{x} \, \mathrm{d}F_C(x),$$

where we used (12) and (15). This shows that $\gamma = \frac{\xi - \gamma}{1 - \gamma} \int_{x \neq 0} \frac{1}{x} \, \mathrm{d}F_C(x)$. We conclude by observing that

$$\theta_{1,h} = \lim_{n \to \infty} \frac{1}{d} \operatorname{trace} \mathbb{E} \left[ (S_{1,n}^\top S_{1,n})^{-1} \right] = \frac{1}{\gamma} \cdot \lim_{n \to \infty} \mathbb{E} \left[ \frac{1}{n} \sum_{i=1}^{d} \frac{1}{\lambda_i} \right] = \frac{1}{\gamma} \int_{x \neq 0} \frac{1}{x} \, \mathrm{d}F_C(x),$$

and consequently, $\theta_{1,h} = \frac{1 - \gamma}{\xi - \gamma}$, which is the claimed result.

2. **Computing $\theta_{2,h}$**

   Unrolling its definition, we have that

$$\theta_{2,h} = \lim_{n \to \infty} \frac{1}{d} \operatorname{trace} \mathbb{E} \left[ (S_{1,n}^\top S_{1,n})^{-2} \right] = \frac{1}{\gamma} \cdot \lim_{n \to \infty} \mathbb{E} \left[ \frac{1}{n} \sum_{i=1}^{d} \frac{1}{\lambda_i^2} \right] = \frac{1}{\gamma} \int_{\{x \neq 0\}} \frac{1}{x^2} \, \mathrm{d}F_C(x),$$

where the limit in the third equation holds and is finite since $F_C$ has support bounded away from 0 on $\mathbb{R}^*$. By definition of $m_C$ and using the fact that $F_C$ has point mass $1 - \gamma$ at 0, we get that

$$\frac{\mathrm{d}m_C(z)}{\mathrm{d}z} = \int \frac{1}{(x - z)^2} \, \mathrm{d}F_C(x) = \frac{1 - \gamma}{z^2} + \int_{\{x \neq 0\}} \frac{1}{(x - z)^2} \, \mathrm{d}F_C(x).$$

Using again the fact that $F_C$ has support bounded away from 0 on $\mathbb{R}^*$ and the dominated convergence theorem, we have that $\gamma\theta_{2,h} = \lim_{z \to 0} \int_{x \neq 0} \frac{1}{(x - z)^2} \, \mathrm{d}F_C(x)$, and thus,

$$\gamma\theta_{2,h} = \lim_{z \to 0} \left\{ \frac{\mathrm{d}m_C(z)}{\mathrm{d}z} - \frac{1 - \gamma}{z^2} \right\}.$$

We denote

$$\triangle := (\gamma + \xi - 2 + z)^2 + 4(z-1)(1-\gamma)(1-\xi)\,,$$

$$\triangle' := \frac{\mathrm{d}\triangle}{\mathrm{d}z} = 2(z + \gamma + \xi - 2) + 4(1-\gamma)(1-\xi)\,.$$

Then, using the expression (4) of $m_C$ and taking the derivative, it follows that

$$\frac{\mathrm{d}m_C(z)}{\mathrm{d}z} - \frac{1-\gamma}{z^2} = \frac{1 - \frac{1}{2\sqrt{\triangle}}(2(z+\gamma+\xi-2) + 4(1-\gamma)(1-\xi))}{2z(1-z)} \tag{16}$$

$$+ \frac{(z+\gamma+\xi-2-\sqrt{\triangle})(2z-1)}{2z^2(z-1)^2} + \frac{\gamma-1}{z^2} \tag{17}$$

$$= \frac{1}{2z^2(z-1)^2}[\triangle_1 + (2\gamma\xi - \gamma - \xi)\triangle_2 - \triangle_3 + \triangle_4], \tag{18}$$

where

$$\begin{cases} \triangle_1 = \frac{z^2(z-1)}{\sqrt{\triangle}} \\ \triangle_2 = \frac{z(z-1)}{\sqrt{\triangle}} \\ \triangle_3 = (2z-1)\sqrt{\triangle} \\ \triangle_4 = z(1-z) + (z+\gamma+\xi-2)(2z-1) + 2(\gamma-1)(z-1)^2. \end{cases}$$

According to L'Hospital rule,

$$\gamma\theta_{2,h} = \lim_{z\to 0} \frac{\triangle_1'' + (2\gamma\xi - \gamma - \xi)\triangle_2'' - \triangle_3'' + \triangle_4''}{2(12z^2 - 12z + 2)} = \lim_{z\to 0} \frac{\triangle_1'' + (2\gamma\xi - \gamma - \xi)\triangle_2'' - \triangle_3'' + \triangle_4''}{4}\,, \tag{19}$$

where $\triangle_i''$ denotes the second derivative of $\triangle_i$ with respect to $z$. After some calculations, we find that

$$\triangle_1''|_{z=0} = -\frac{2}{\xi - \gamma}\,,$$

$$\triangle_2''|_{z=0} = \frac{2}{\xi - \gamma} + \frac{4\gamma\xi - 2\gamma - 2\xi}{(\xi - \gamma)^3}\,,$$

$$\triangle_3''|_{z=0} = \frac{4(2\gamma\xi - \gamma - \xi) - 1}{\xi - \gamma} + \frac{(2\gamma\xi - \gamma - \xi)^2}{(\xi - \gamma)^3}\,,$$

$$\triangle_4''|_{z=0} = 2(2\gamma - 1)\,.$$

Using (19), it follows that

$$\gamma\theta_{2,h} = \frac{1}{4}\left(\frac{-(2\gamma-1)^2}{\xi-\gamma} + \frac{(2\gamma\xi - \gamma - \xi)^2}{(\xi-\gamma)^3}\right) = \frac{\gamma(1-\gamma)(\gamma^2 + \xi - 2\gamma\xi)}{(\xi - \gamma)^3}\,,$$

and finally, we obtain the claimed expression, that is, $\theta_{2,h} = \frac{(1-\gamma)(\gamma^2 + \xi - 2\gamma\xi)}{(\xi - \gamma)^3}$.

## A.2  Proof of Theorem 3.1

*Proof.* Let $\{S_t\}$ be a sequence of independent $m \times n$ Haar matrices, and let $\{x_t\}$ be the sequence of iterates generated by the update (2) with $\mu_t = \theta_{1,h}/\theta_{2,h}$ and $\beta_t = 0$. Recall that we denote $\Delta_t = U^\top A(x_t - x^*)$, where $A = U\Sigma V^\top$ is a thin singular value decomposition of $A$. For $t \geqslant 0$, we have that

$$A\left(A^\top S^\top SA\right)^{-1} A^\top = U\Sigma V^\top \left(V\Sigma U^\top S^\top SU\Sigma V^\top\right)^{-1} V\Sigma U^\top$$

$$= U\Sigma V^\top V\Sigma^{-1}(U^\top S^\top SU)^{-1}\Sigma^{-1}VV^\top \Sigma U^\top$$

$$= U(U^\top S^\top SU)^{-1}U^\top$$

Multiplying both sides of the update formula (2) by $A$, subtracting $Ax^*$ and using the normal equation $A^\top A x^* = A^\top b$, we find that

$$A(x_{t+1} - x^*) = \left(I_n - \mu_t U(U^\top S_t^\top S_t U)^{-1}U^\top\right) A(x_t - x^*). \tag{20}$$

Multiplying both sides of (20) by $U^\top$, using the definition of $\Delta_t$ and the fact that $U^\top U = I_d$, it follows that

$$\begin{aligned}
\Delta_{t+1} &= U^\top \left(I_n - \mu_t U(U^\top S_t^\top S_t U)^{-1}U^\top\right) A(x_t - x^*) \\
&= \left(U^\top - \mu_t U^\top U(U^\top S_t^\top S_t U)^{-1}U^\top\right)(Ax_t - x^*) \\
&= \left(I_d - \mu_t(U^\top S_t^\top S_t U)^{-1}\right)\Delta_t,
\end{aligned}$$

and then, taking the squared norm,

$$\|\Delta_{t+1}\|^2 = \Delta_t^\top \left(I_d - \mu_t(U^\top S_t^\top S_t U)^{-1}\right)^2 \Delta_t.$$

Taking the expectation with respect to $S_t$ and using the independence of $S_t$ with respect to $S_0, \ldots, S_{t-1}$, we obtain that

$$\mathbb{E}_{S_t}\left[\|\Delta_{t+1}\|^2\right] = \Delta_t^\top \mathbb{E}\left[\left(I_d - \mu_t(U^\top S_t^\top S_t U)^{-1}\right)^2\right]\Delta_t \tag{21}$$

$$= \Delta_t^\top \left(I_d - 2\mu_t \,\mathbb{E}\left[(U^\top S_t^\top S_t U)^{-1}\right] + \mu_t^2\,\mathbb{E}\left[(U^\top S_t^\top S_t U)^{-2}\right]\right)\Delta_t. \tag{22}$$

We write the spectral decomposition $U^\top S_t^\top S_t U = V\Sigma V^\top$ where $\Sigma$ is diagonal with positive entries $\lambda_1, \ldots, \lambda_d$ and $V_t = [v_1, \ldots, v_d]$ is a $d \times d$ orthogonal matrix. The matrix $S_t U$ is distributed as the $m \times d$ upper-left block of an $n \times n$ Haar matrix. Therefore, $S_t U$ is right rotationally invariant, and so is the matrix $V$. It follows that $\lambda_i v_{ik} v_{i\ell} \overset{\mathrm{d}}{=} -\lambda_i v_{ik} v_{i\ell}$ for any index $i$ and any indices $k \neq \ell$. Then, for any $p \in \{1, 2\}$ and any $k \neq \ell$, we have

$$\mathbb{E}\left[\left((U^\top S^\top SU)^{-p}\right)_{k\ell}\right] = \sum_{i=1}^d \mathbb{E}\left[\lambda_i^{-p} v_{ik} v_{i\ell}\right] = -\sum_{i=1}^d \mathbb{E}\left[\lambda_i^{-p} v_{ik} v_{i\ell}\right],$$

which implies that the off-diagonal term $\mathbb{E}\left[\left((U^\top S^\top SU)^{-p}\right)_{k\ell}\right]$ is equal to $0$. Further, by permutation invariance of the matrix $V$, we get that for any $k$,

$$\mathbb{E}\left[\left((U^\top S^\top SU)^{-p}\right)_{kk}\right] = \frac{1}{d}\operatorname{trace}\mathbb{E}\left[(U^\top S^\top SU)^{-p}\right],$$

or equivalently, $\mathbb{E}\left[(U^\top S^\top SU)^{-p}\right] = \theta_{p,n} I_d$ where $\theta_{p,n} := d^{-1}\operatorname{trace}\mathbb{E}\left[(U^\top S^\top SU)^{-p}\right]$. Then, using (22), it follows that

$$\begin{aligned}
\mathbb{E}_{S_t}\left[\|\Delta_{t+1}\|^2\right] &= \Delta_t^\top \left(I_d - 2\mu_t\,\theta_{1,n}I_d + \mu_t^2\,\theta_{2,n}I_d\right)\Delta_t \\
&= (1 - 2\mu_t\theta_{1,n} + \mu_t^2\theta_{2,n})\cdot\|\Delta_t\|^2 \\
&= \left(1 - \frac{\theta_{1,n}^2}{\theta_{2,n}} + \left(\frac{\theta_{1,n}}{\sqrt{\theta_{2,n}}} - \mu_t\sqrt{\theta_{2,n}}\right)^2\right)\cdot\|\Delta_t\|^2.
\end{aligned}$$

By induction, we further obtain

$$\frac{\mathbb{E}\left[\|\Delta_t\|^2\right]}{\|\Delta_0\|^2} = \prod_{j=0}^{t-1}\left(1 - \frac{\theta_{1,n}^2}{\theta_{2,n}} + \left(\frac{\theta_{1,n}}{\sqrt{\theta_{2,n}}} - \mu_j\sqrt{\theta_{2,n}}\right)^2\right).$$

Taking the limit $n \to \infty$ and using the definition $\theta_{h,p} = \lim_{n\to\infty}\theta_{p,n}$ for $p \in \{1, 2\}$, we find that

$$\lim_{n\to\infty}\frac{\mathbb{E}\left[\|\Delta_t\|^2\right]}{\|\Delta_0\|^2} = \prod_{j=0}^{t-1}\left(1 - \frac{\theta_{1,h}^2}{\theta_{2,h}} + \left(\frac{\theta_{1,h}}{\sqrt{\theta_{2,h}}} - \mu_j\sqrt{\theta_{2,h}}\right)^2\right).$$

The above right-hand side is minimized at $\mu_j = \theta_{1,h}/\theta_{2,h}$ for all times steps $j \geqslant 0$, which yields the error formula

$$\lim_{n \to \infty} \frac{\mathbb{E}\left[\|\Delta_t\|^2\right]}{\|\Delta_0\|^2} = \left(1 - \frac{\theta_{1,h}^2}{\theta_{2,h}}\right)^t.$$

Plugging-in the expressions of $\theta_{1,h}$ and $\theta_{2,h}$, we obtain the claimed convergence rate $\rho_h$.

It remains to prove that $\rho_h$ is the best rate one may achieve with the update (2) along with Haar embeddings. It is actually an immediate consequence of Theorem 2 in [4] whose assumptions (precisely, Assumption 1 in [4]) are trivially satisfied by Haar embeddings.

$\square$

## A.3   Calculations of $\theta_{1,h}$ and $\theta_{2,h}$ for SRHT

Our analysis proceeds in a way similar to the analysis of the Haar case, and we describe in this paragraph the main steps. Denote by $F_S$ the l.s.d. of $U^\top S^\top S U$ and by $F_{S,n}$ its e.s.d. As we did for the Haar case with the matrix $C_n$, we introduce here an auxiliary matrix $G_n$ whose e.s.d. is related to $F_{S,n}$. Then, we characterize the $\eta$-transform $\eta_G$ of its l.s.d. $F_G$. Our analysis for $\eta_G$ uses recent results on *asymptotically liberating sequences* from free probability [1]. This technique has also been used in the prior work [3]. Finally, we show that $\eta_G$ is equal to the $\eta$-transform $\eta_C$ of $F_C$, and we conclude that $F_S = F_h$.

Let $S = BH_nDP$ be the $n \times n$ SRHT matrix (before discarding the rows) as defined in Section 4 in the paper, and $U$ be an $n \times d$ deterministic matrix with orthonormal columns. Note that whether we consider the zero rows or not in the matrix $S$, the matrix $U^\top S^\top S U$ remains the same, and so does its l.s.d. The matrices $B$, $H_n$ and $D$ are all symmetric matrices, and they respectively satisfy $B^2 = B$, $H_n^2 = I_n$ and $D^2 = I_n$, and $P$ is also an orthogonal matrix. Then, we have that $S^\top S = P^\top DH_nBH_nDP$, and further,

$$(S^\top S)^2 = P^\top DH_nBH_nDPP^\top DH_nBH_nDP = P^\top DH_nBH_nDP = S^\top S.$$

We first have the following observation, whose proof is deferred to Appendix B.3.

**Lemma A.3.** *For $P$, $B$, $D$, $H_n$ and $U$ defined as above, we have the following equality in distribution*

$$U^\top(P^\top DH_n)B(HDP)U \stackrel{\mathrm{d}}{=} U^\top(P^\top DH_nDP)B(P^\top DH_nDP)U. \tag{23}$$

We now proceed with asymptotic statements, and we introduce the subscript $n$ to all matrices. We set $W_n := P_n^\top D_nH_nD_nP_n$. It holds that the matrix $U_n^\top W_nB_nW_nU_n$ has the same nonzero eigenvalues as $G_n := B_nW_nU_nU_n^\top W_nB_n$, so that we first find the l.s.d. of the matrix $G_n$. The reader may notice that $G_n$ plays a similar role in the analysis of the SRHT case, to that of the matrix $C_n$ in the analysis of the Haar case.

The following result states the asymptotic freeness of the matrices $B_n$ and $W_nU_nU_n^\top W_n$. Its proof follows directly from Corollaries 3.5 and 3.7 by [1].

**Lemma A.4.** *Let $B_n, W_n, U_n$ be defined as above. Then, the matrices $\{B_n, W_nU_nU_n^\top W_n\}$ are asymptotically free in the limit of the non-commutative probability spaces of random matrices. Consequently, the e.s.d. of the matrix $G_n = B_nW_nU_nU_n^\top W_nB_n$ converges to the freely multiplicative convolution of the l.s.d. $F_B$ of $B_n$ and the l.s.d. $F_U$ of $U_nU_n^\top$, that is, $G_n$ has l.s.d. given by $F_G = F_B \boxtimes F_U$.*

Since the density of the l.s.d. $F_B$ is $f_B = \xi\delta_1 + (1-\xi)\delta_0$ and and the density of $F_U$ is $f_U = \gamma\delta_1 + (1-\gamma)\delta_0$, we have that the $S$-transforms $S_B$ of $F_B$ and $S_U$ of $F_U$ are respectively equal to $S_B(y) = \frac{y+1}{y+\xi}$ and $S_U(y) = \frac{y+1}{y+\gamma}$. From Lemma A.4, it follows that the $S$-transform $S_G$ of $F_G$ is the product of $S_B$ and $S_U$, i.e.,

$$S_G(y) = S_U(y)S_B(y) = \frac{(y+1)^2}{(y+\xi)(y+\gamma)}. \tag{24}$$

First, note that using their respective definitions, the $S$-transform of $F_G$ and its $\eta$-transform $\eta_G$ are related by the equation $\eta_G\left(-\frac{y}{y+1}S_G(y)\right) = y+1$. Plugging-in the expression (24) of $S_G(y)$ into the latter equation, we obtain that

$$\eta_G\left(-\frac{y(y+1)}{(y+\gamma)(y+\xi)}\right) = y+1\,.$$

Letting $z = -\frac{(y+\gamma)(y+\xi)}{y(y+1)}$ and using the relationship (8) between the Stieltjes and $\eta$-transforms, we find that the Stieltjes transform $m_G$ of $G$ is equal to

$$m_G(z) = \frac{z+\gamma+\xi-2-\sqrt{g(z)}}{2z(1-z)}\,,$$

where $g(z) = (\gamma+\xi-2+z)^2 + 4(z-1)(1-\gamma)(1-\xi)$. Hence, we get that $m_G(z) = m_C(z)$, that is, $F_G = F_C$.

Further, the matrix $G_n$ has the same non-zero eigenvalues as the matrix $U_n^\top W_n B_n W_n U_n$ which, according to Lemma A.3, is equal in distribution to $U_n^\top S_n^\top S_n U_n$. Denote by $\lambda_1, \ldots, \lambda_{\widetilde{d}}$ the non-zero eigenvalues of $U_n^\top S_n^\top S_n U_n$, where $\widetilde{d}$ is itself a random number due to the randomness of non-zero rows $\widetilde{m}$. Hence, the e.s.d $F_{G,n}$ of $G_n$ and the e.s.d. $F_{S,n}$ of $U_n^\top S_n^\top S_n U_n$ satisfy (see Appendix B.4)

$$F_{G_n}(x) \stackrel{\mathrm{d}}{=} \left(1-\frac{d}{n}\right)\mathbf{1}_{\{x\geqslant 0\}} + \frac{d}{n}F_{S,n}(x)\,. \tag{25}$$

Thus, we obtain that $F_{S,n}$ converges weakly almost surely to the distribution

$$F_S(x) := \frac{1}{\gamma}\left(F_G(x) - (1-\gamma)\mathbf{1}_{\{x\geqslant 0\}}\right) = \frac{1}{\gamma}\left(F_C(x) - (1-\gamma)\mathbf{1}_{\{x\geqslant 0\}}\right)\,. \tag{26}$$

The latter expression is equal to $F_h(x)$ according to (6), so that $F_S(x) = F_h(x)$. The analysis of the traces of the expected first and second inverse moments only involves the limiting distribution (we refer the reader to the proof of the expressions of $\theta_{1,h}$ and $\theta_{2,h}$, in Section A.1). Due to the equality $F_h = F_S$, they remain the same with SRHT matrices, which concludes the proof of Lemma 4.3.

## A.4 Proof of Theorem 4.1 and 4.2

Let $\{S_t\}$ be a sequence of independent $m \times n$ SRHT matrices, and let $\{x_t\}$ be the sequence of iterates generated by the update (2) with $\mu_t = \theta_{1,h}/\theta_{2,h}$ and $\beta_t = 0$. Denote $\Delta_t = U^\top A(x_t - x^*)$ the sequence of error vectors. The proof follows exactly the same lines as for Theorem 4.1 up to the relationship (22), which we recall here,

$$\mathbb{E}_{S_t}\left[\|\Delta_{t+1}\|^2\right] = \mathbb{E}_{S_t}\left[\Delta_t^\top\left(I_d - \mu_t\left(U^\top S_t^\top S_t U\right)^{-1}\right)^2\Delta_t\right]\,. \tag{27}$$

Denote $Q_t = I_d - \mu_t\left(U^\top S_t^\top S_t U\right)^{-1}$. It holds that $\Delta_{t+1} = Q_t\Delta_t$ as previously shown. Hence, by induction, we obtain that

$$\mathbb{E}\left[\|\Delta_t\|^2\right] = \operatorname{trace}\mathbb{E}\left[Q_0\ldots Q_{t-1}Q_{t-1}\ldots Q_0\Delta_0\Delta_0^\top\right]\,. \tag{28}$$

Using the independence of $\Delta_0$ and the $Q_i$, and the assumption $\mathbb{E}\left[\Delta_0\Delta_0^\top\right] = I_d/d$, it follows that

$$\mathbb{E}\left[\|\Delta_t\|^2\right] = \frac{1}{d}\operatorname{trace}\mathbb{E}\left[Q_1\ldots Q_{t-1}Q_{t-1}\ldots Q_0^2\right]\,. \tag{29}$$

It holds that the matrix $Q_0^2$ is asymptotically free from $Q_{t-1}\ldots Q_1$. Therefore, using the trace decoupling relation (7), we have that

$$\begin{aligned}
\lim_{n\to\infty}\mathbb{E}\left[\|\Delta_t\|^2\right] &= \lim_{n\to\infty}\frac{1}{d}\operatorname{trace}\mathbb{E}\left[Q_1\ldots Q_{t-1}Q_{t-1}\ldots Q_0^2\right] \\
&= \lim_{n\to\infty}\frac{1}{d}\operatorname{trace}\mathbb{E}\left[Q_0^2\right]\cdot\lim_{n\to\infty}\frac{1}{d}\operatorname{trace}\mathbb{E}\left[Q_2\ldots Q_{t-1}Q_{t-1}\cdots Q_1^2\right]\,.
\end{aligned}$$

Note that $\lim_{n\to\infty} \frac{1}{d} \operatorname{trace} \mathbb{E}\left[Q_0^2\right] = (1 - 2\mu_0\theta_{1,h} + \mu_0^2\theta_{2,h})$. Repeating the same asymptotic freeness argument between $Q_1^2$ and $Q_{t-1}\ldots Q_2$ and plugging-in $\mu_j = \theta_{1,h}/\theta_{2,h}$, we finally obtain the claimed result,

$$
\lim_{n\to\infty} \mathbb{E}\left[\|\Delta_{t+1}\|^2\right] = \prod_{j=0}^{t-1} \left(1 - \mu_j\theta_{1,h} + \mu_j^2\theta_{2,h}\right)
$$
$$
= \left(1 - \frac{\theta_{1,h}^2}{\theta_{2,h}}\right)^t .
$$

The proof of Theorem 4.2 immediately follows from an alternative upper-bound on the expression (28) for the norm of the error. In particular, we note that

$$
\mathbb{E}\left[\|\Delta_t\|^2\right] = \operatorname{trace} \mathbb{E}\left[Q_0\ldots Q_{t-1}Q_{t-1}\ldots Q_0\Delta_0\Delta_0^\top\right]
$$
$$
\leq \|\Delta_0\Delta_0^\top\|_2 \operatorname{trace} \mathbb{E}\left[Q_0\ldots Q_{t-1}Q_{t-1}\ldots Q_0\right]
$$
$$
= d\|\Delta_0\|_2^2 \frac{1}{d} \operatorname{trace} \mathbb{E}\left[Q_0\ldots Q_{t-1}Q_{t-1}\ldots Q_0\right] .
$$

We then combine the earlier expression (29) with the above upper-bound and complete the proof.

**Remark A.1.** *In view of equations (4-6) in [1], one can show that asymptotic freeness between $U^\top S^\top S U$ and a rank-one matrix $vv^\top$ holds provided that $\|v\|_2 < \infty$ as the dimensions grow to infinity. One could then wonder whether such a result can be applied to our setting, in order to remove the assumption $\mathbb{E}\Delta_0\Delta_0^\top = \frac{1}{d} \cdot I_d$. Using (28), dividing by $\mathbb{E}\|\Delta_0\|^2$ and denoting $\widetilde{\Delta}_0 = \frac{\Delta_0}{\sqrt{\mathbb{E}\|\Delta_0\|^2/d}}$, we get*

$$
\frac{\mathbb{E}\|\Delta_t\|^2}{\mathbb{E}\|\Delta_0\|^2} = \frac{1}{d} \operatorname{trace} \mathbb{E}\left[Q_0\ldots Q_{t-1}Q_{t-1}\ldots Q_0\widetilde{\Delta}_0\widetilde{\Delta}_0^\top\right] .
$$

*Provided we have asymptotic freeness between $\widetilde{\Delta}_0\widetilde{\Delta}_0^\top$ and $Q_0\ldots Q_{t-1}Q_{t-1}\ldots Q_0$, then we have*

$$
\lim_{n\to\infty} \frac{\mathbb{E}\|\Delta_t\|^2}{\mathbb{E}\|\Delta_0\|^2} = \lim_{n\infty} \frac{1}{d} \operatorname{trace} \mathbb{E}\left[Q_0\ldots Q_{t-1}Q_{t-1}\ldots Q_0\right] \cdot \lim_{n\infty} \frac{1}{d} \operatorname{trace} \mathbb{E}\left[\widetilde{\Delta}_0\widetilde{\Delta}_0^\top\right]
$$

*According to our previous analysis, the term $\lim_{n\infty} \frac{1}{d} \operatorname{trace} \mathbb{E}\left[Q_0\ldots Q_{t-1}Q_{t-1}\ldots Q_0\right]$ is equal to $(1 - \frac{\theta_{1,h}^2}{\theta_{2,h}})^t$. On the other hand, the term $\lim_{n\infty} \frac{1}{d} \operatorname{trace} \mathbb{E}\left[\widetilde{\Delta}_0\widetilde{\Delta}_0^\top\right]$ is equal to 1, so that we would get the claimed result. But, for asymptotic freeness to hold between $\widetilde{\Delta}_0\widetilde{\Delta}_0^\top$ and $Q_0\ldots Q_{t-1}Q_{t-1}\ldots Q_0$, we need $\|\widetilde{\Delta}_0\| < \infty$, and this assumption seems too strong: for instance, if $\Delta_0$ is deterministic, then $\|\widetilde{\Delta}_0\| = \sqrt{d}$ which is unbounded as the dimensions grow to infinity.*

# B Proofs of the auxiliary results

## B.1 Proof of the bounds on the support of $F_h$ (Lemma A.2)

*Proof.* We show that the support of $F_h$ satisfies

$$
\inf \operatorname{supp}(F_h) \geqslant \frac{\left(1 - \sqrt{\rho_g}\right)^2}{(1 + \frac{1}{\sqrt{\xi}})^2} .
$$

Let $S$ be an $m \times n$ Haar matrix, $U$ an $n \times d$ deterministic matrix with orthonormal columns, and $S_g$ be an $m \times n$ matrix independent of $S$, with i.i.d. entries $\mathcal{N}(0, 1/m)$. Write $S_g = \Omega_\ell\Sigma\Omega_r$ a singular value decomposition of $S_g$. It

holds that $\Omega_\ell$ is an $m \times m$ Haar matrix, independent of the $m \times m$ diagonal matrix of singular values $\Sigma$, and $\Omega_r \overset{\mathrm{d}}{=} S$, so that $\Omega_\ell \Sigma S \overset{\mathrm{d}}{=} S_g$. Further, the operator norm of $\Sigma$ satisfies $\lim_{n\to\infty} \|\Sigma\|_2 = \left(1 + \frac{1}{\sqrt{\xi}}\right)$ almost surely. Then,

$$\sigma_{\min}(SU) = \min_{\|x\|=1} \|SUx\| \geqslant \min_{\|x\|=1} \frac{\|\Sigma SUx\|}{\|\Sigma\|_2}$$
$$= \frac{1}{\|\Sigma\|_2} \cdot \min_{\|x\|=1} \|\Omega_\ell \Sigma SUx\|\,.$$

Almost surely, $\min_{\|x\|=1} \|\Omega_\ell \Sigma Sx\| \to (1 - \sqrt{\rho_g})$ as $n \to \infty$. Thus, almost surely, $\liminf_{n\to\infty} \sigma_{\min}(SU) \geqslant \frac{(1-\sqrt{\rho_g})}{(1+\frac{1}{\sqrt{\xi}})}$, which yields the claimed lower bound on the support of $F_h$. $\quad\square$

## B.2  Characterization of the e.s.d. of $C_n$

Recall the definition (3) of the matrix $C_n$,

$$C_n = \begin{pmatrix} I_m & 0 \\ 0 & 0 \end{pmatrix} \mathbb{W}_n \begin{pmatrix} I_d & 0 \\ 0 & 0 \end{pmatrix} \mathbb{W}_n^\top \begin{pmatrix} I_m & 0 \\ 0 & 0 \end{pmatrix}\,.$$

We leverage Theorem 4.11 from [2], which we recall for the sake of completeness.

**Theorem B.1** (Theorem 4.11, [2]). *Let $D_n \in \mathbb{R}^{n\times n}$ and $T_n \in \mathbb{R}^{n\times n}$ be diagonal non-negative matrices, and $\mathbb{W}_n \in \mathbb{R}^{n\times n}$ be a Haar matrix. Denote $F_D$ and $F_T$ the respective l.s.d. of $D_n$ and $T_n$. Denote $C_n$ the matrix $C_n := D_n^{\frac{1}{2}} \mathbb{W}_n T_n \mathbb{W}_n^\top D_n^{\frac{1}{2}}$. Then, as $n$ tends to infinity, the e.s.d. of $C_n$ converges to $F$ whose $\eta$-transform $\eta_F$ satisfies*

$$\eta_F(z) = \int \frac{1}{z\gamma(z)x + 1} \, \mathrm{d}F_D(x)\,,$$
$$\gamma(z) = \int \frac{x}{\eta_F(z) + z\delta(z)x} \, \mathrm{d}F_T(x)\,,$$
$$\delta(z) = \int \frac{x}{z\gamma(z)x + 1} \mathrm{d}F_D(x)\,.$$

The e.s.d. of $\begin{pmatrix} I_d & 0 \\ 0 & 0 \end{pmatrix}$ converges to the distribution $F_\gamma$ with density $\gamma\delta_1 + (1-\gamma)\delta_0$, and the e.s.d. of $\begin{pmatrix} I_m & 0 \\ 0 & 0 \end{pmatrix}$ converges to the distribution $F_\xi$ with density $\xi\delta_1 + (1-\xi)\delta_0$. Then, according to Theorem B.1, the e.s.d. of $C_n$ converges to a distribution $F_C$, whose $\eta$-transform $\eta_C$ is solution of the following system of equations,

$$\eta_C(z) = \int \frac{1}{z\gamma(z)x + 1} \, \mathrm{d}F_\xi(x)\,, \tag{30}$$
$$\gamma(z) = \int \frac{x}{\eta_C(z) + z\delta(z)x} \, \mathrm{d}F_\gamma(x)\,, \tag{31}$$
$$\delta(z) = \int \frac{x}{z\gamma(z)x + 1} \, \mathrm{d}F_\xi(x)\,. \tag{32}$$

Plugging the above expressions of $F_\xi$ and $F_\gamma$ into the above equations, and after simplification, we obtain that $\eta_C$ is solution of the following second-order equation

$$\eta_C(z) = (1 - \gamma) + \frac{\gamma}{1 + z\left(1 + \frac{\xi - 1}{\eta_C(z)}\right)}\,, \tag{33}$$

Plugging the relationship (8) between the Stieltjes and $\eta$-transforms into (33), we find that

$$m_C(z) = \frac{z + \gamma + \xi - 2 - \sqrt{g(z)}}{2z(1 - z)}\,, \tag{34}$$

where $g(z) = (\gamma + \xi - 2 + z)^2 + 4(z - 1)(1 - \gamma)(1 - \xi)$, and we choose the branch of the square-root such that $m_C(z) \in \mathbb{C}^+$ for $z \in \mathbb{C}^+$, $m_C(z) \in \mathbb{C}^-$ for $z \in \mathbb{C}^-$ and $m_C(z) > 0$ for $z < 0$.

## B.3 Proof of Lemma A.3

*Proof.* Note that both $B$ and $D$ are diagonal matrices whose diagonal entries are i.i.d. random variables, and $P$ is a permutation matrix. Define $\tilde{B} = PBP^\top$ and $\tilde{D} = P^\top DP$, then we have

$$\tilde{B} \overset{d}{=} B, \quad \tilde{D} \overset{d}{=} D$$

and

$$DP = P\tilde{D}, \quad P^\top D = \tilde{D}P^\top. \tag{35}$$

It follows that

$$
\begin{aligned}
U^\top P^\top DH_n DPBP^\top DH_n DPU &= U^\top P^\top DH_n P\tilde{D}B\tilde{D}P^\top H_n DPU \\
&= U^\top P^\top DH_n PB\tilde{D}^2 P^\top H_n DPU \\
&= U^\top P^\top DH_n PBP^\top H_n DPU \\
&= U^\top P^\top DH_n \tilde{B} H_n DPU \\
&\overset{d}{=} U^\top P^\top DH_n BH_n DPU,
\end{aligned}
$$

where the first equation follows from (35), the second equation holds because $\tilde{D}$ and $B$ are diagonal so they commute, while the third equation holds because $\tilde{D}^2 = I_n$. $\qquad\square$

## B.4 Proof of the identity (25)

We note that

$$
\begin{aligned}
F_{G_n}(x) &\overset{d}{=} \left(1 - \frac{\tilde{d}}{n}\right) \mathbf{1}_{\{x \geqslant 0\}} + \frac{1}{n} \sum_{j=1}^{\tilde{d}} \mathbf{1}_{\{x \geqslant \lambda_j\}} \\
&= \left(1 - \frac{\tilde{d}}{n}\right) \mathbf{1}_{\{x \geqslant 0\}} + \frac{d}{n} \cdot \frac{1}{d} \sum_{j=1}^{\tilde{d}} \mathbf{1}_{\{x \geqslant \lambda_j\}} \\
&= \left(1 - \frac{\tilde{d}}{n}\right) \mathbf{1}_{\{x \geqslant 0\}} + \frac{d}{n} \left( F_{S,n}(x) - \left(\frac{d - \tilde{d}}{d}\right) \mathbf{1}_{\{x \geqslant 0\}} \right) \\
&= \left(1 - \frac{d}{n}\right) \mathbf{1}_{\{x \geqslant 0\}} + \frac{d}{n} F_{S,n}(x),
\end{aligned}
$$

which proves (25).