[Reviews · NeurIPS 2020]

Review 1

Summary and Contributions: This paper analyzes overconstrained L2 regression problem based on randomized Hessian sketches, particularly using SRHT or Haar matrix embedding. The method relies on pre-conditioned heavy-ball updates; the use SRHT or Haar embedding improves upon Gaussian projection and achieves asymptotic optimality in terms of the step size as well as convergence rate.

Strengths: Apart from the main results, the authors also come up with explicit formulations for first two inverse moments of SRHT or Haar sketches, which may be of independent interest. However, I have some comments/questions given below.

Weaknesses: 1. There is the following recent paper that came up with optimal first order methods for the same overconstrained L2 regression: https://proceedings.icml.cc/static/paper_files/icml/2020/3430-Paper.pdf. In the above paper, the authors used a fixed sketch (in contrast to the current work in which authors used different sketching matrices generated independently in each iteration) which achieved similar or better convergence rate than this paper. The above paper was based on a larger class of pre-conditioned quasi-Newton like methods. While the analyses of these two papers are somewhat different from each other, I am wondering why would one care about the variable sketches (i.e. the current work), rather than the fixed sketch (i.e. the paper mentioned above) that performs better both in terms of the convergence rate and the computation (as one has to generate S only once)? 2. Can this analysis be also extended to accommodate count-sketch or any other sparse sketching matrices ?

Correctness: The theoretical claims as well as the experimental evaluations seem correct

Clarity: The paper is well-organized and easy to follow.

Relation to Prior Work: The authors missed the comparison with one of the recent papers that I mentioned in weaknesses.

Reproducibility: Yes

Additional Feedback: -------------------------------------------------Post rebuttal-------------------------------------------- I thank the authors for addressing my concerns. After going through other reviews and the authors feedback, I further increase my score. In the revised version, it will be great if the authors could do a thorough comparison with the papers that R3 and I mentioned in the review. --------------------------------------------------------------------------------------------------------------


Review 2

Summary and Contributions: This paper aims at solving the over-determined least-squares problem with sketching randomized methods. By extending results based on the limiting spectrum of matrices (lsd), the authors are able to give a novel closed form formula for the inverse of projection matrix. This leads to an analysis of the Iterative Hessian Sketch (IHS) algorithm for truncated Haar and Subsampled Randomized Hadamard (SRH) transforms. For this algorithm, authors give optimal step sizes and corresponding convergence rate. This paper highlights the similarity between the spectrum of the Haar and randomized Hadamard, which explains that they lead to the same convergence rate, which is better than for Gaussian projections.

Strengths: The key contributions of the paper are the following - 1) give an optimal version of IHS, that is optimal values for the step sizes $\mu_t$ and the momentum parameters $\beta_t$ - 2) proof of a gain of magnitude $log (d)$ compared to the complexity of state-of-the-art algorithms, like gradient descent (GD), conjugate gradient (CG) and precondition-CG - 3) Derivation of a formula for $\theta_{2,h}$, the second moment of for Haar/SRHT - 4) derivation of asymptotic analysis of the spectrum of $U^\top S^\top S U$ to explain the better performance of SRHT compared to random Gaussian projections

Weaknesses: - 1) Even if the complexity of the algorithm presented in this paper gains a factor log(d), they seem that it might be slower in some cases because the epsilon term cannot really be treated as a "constant". For instance, if d is small and the required precision epsilon is large, and since the terms in the complexity C_n eq. (18) are all multiplied by log(1/epsilon), it might happen that it would be slower than classical competitors. - 2) Unfortunately, the optimality of IHS is only proved for Haar transform. We wish we could have a lower-bound for the error in the SRHT case, like in eq. (11) instead of just eq. (14) - 3) The SRHT used in this paper is not exactly the original method because it incorporates an additional permutation step with matrix P in line 235 - 4) No numerical comparisons against state-of-the-art randomized methods are given. This is sorely lack in the numerical experiments section. Could you please argue on this?

Correctness: - 1) Claims are correct and the proved and unproved elements are clearly distinguished from one another (like the non-optimality of the convergence rate for SRHT). Proofs are conscientiously sketch in the main paper, which makes the reading easy, and well detailed in the appendix. - 2) Yet, numerical experiments are not detailed enough. No detail is given about how the Hadamard transform is performed for instance. Through matrix multiplication with Hadamard matrix or Fast Walsh-Hadamard Transform (FWHT)?. How to deal with n which is not a power of 2 for SRHT ? - 3) Authors only give plots as functions of the number of iterations, yet applying a SRHT, Haar or Gaussian sketch does not take the same number of computations.

Clarity: As mentioned above, the paper is very clear.

Relation to Prior Work: The introduction clearly sums up previous work and the theoretical improvements proven in this article.

Reproducibility: No

Additional Feedback: Please find below a list of questions and comments : - 1) Did you experiment applying the truncated Walsh-Hadamard transform (Ailon & Liberty. Fast dimension reduction using Rademacher series on dual BCH codes. In Discrete & Computational Geometry, 2009.) when using SRHT ? - 2) lines 199-207 Could you comment on the fact that m is constrained to be greater than d? Would it be possible to achieve better performances with smaller m? Is there any theory about this? - 3) Could you please precisely give the references claiming that $m \approx d \log(d)$ is prescribed for state-of-the-art algorithms and for which algorithm? - 4) Below Theorem 4.1, there is an explanation on why this additional assumption $\mathbb{E} \[ \Delta_0 \Delta_0^\top \]$ = (1/d) I_d$ is a "mild assumption". I did not understood the provided argument. Is this assumption often/easily met? - 5) lines 222-223 and 264-265 it is mentioned that "SRHT [...] contains less randomness, but is more structured and faster to generate" than Haar matrix. Could you please detail this? - 6) line 229 A comment on how to deal in the case where n is not a power of 2, with padding, subsampling or other methods, would be great. Especially if such preprocessing methods degrade the performance. - 7) lines 231-238 : Could you please detail here why such a modification of classical SRHT, with random row permutations, is required? - 8) Figure 1 & 2 are almost the same, putting only Figure 2 is enough. Another comment on this Figure 2: it is very hard to distinguish between the two colors. Changing one of them would be great. - 9) Figure 3 : are there error bars? One can barely see the dotted lines. Maybe making the font bigger and putting more space between two dots would be better. - 10) lines 293-295 and Appendix C: Int standard machine learning datasets n is not a power of 2. How did you deal with this for SRHT? (redundant question) - 11) lines 302-303 : I found it strange to decompose IHS and pCG the same way since preconditioning is only applied once. I found this sentend confusing. Maybe it could be reformulate. - 12) Appendix C: "For SRHT, we use the optimal step sizes". I thought it was not proven to be optimal for SRHT. Did I miss something? - 13) Will the code be available online? Comments on notations and typos: - 14) line 153: S is $m \times d$ whereas sketch matrix is $m \times n$ in all other parts of the paper. This might be confusing. - 15) line 185: typo in the Stieltjes transform $m(x) = ...$ -> $m(z) = ...$ - 16) line 264; typo "although" - 17) line 380: "J. Tropp [...] HAdamard transform [...]" Nitpicking: - 18) If there is room left at the end, a summary of remaining open questions (non optimality of the rate for SRHT, etc) raised in the core of the paper would be great. ====== EDIT AFTER REBUTTAL ====== The fact that in this paper, you must refresh the sketch every iteration, should be clearly compared to the papers https://arxiv.org/pdf/2002.09488.pdf and https://arxiv.org/abs/2006.05874. I guess that the practical improvement of this paper is not its major point. You should clearly state (like in your feedbacks) that despite having "only" asymptotic results, your analysis is still interesting since traditional sketching theory has worked on finite-sample guarantees, which can hide large constants. Maybe you could put even more stress on the technical aspects of the paper, eg the asymptotic analysis, that might make an impact on the community. Other reviewers mentioned that it seems you are overselling some of your results, please try to find a way to take this remark into account.


Review 3

Summary and Contributions: The paper provides improved analysis of iterative Hessian sketching with Gaussian, Haar (random orthonormal), and subsampled randomized Hadamard (SRHT) sketching matrix. It proves tight bounds for their rate of convergence and shows that both Haar and SRHT converge faster than Gaussian sketches. The sophisticated asymptotic analysis is based on recent results in random matrix theory.

Strengths: - Sophisticated and tight theoretical analysis. - Closed formula for theoretically optimal step size and momentum. - Experimental evaluation with synthetic data, MNIST, and CIFAR data sets support the theoretical claims.

Weaknesses: - Analysis is asymptotic and holds in the infinite limit only. - Paper assumes that ratio of variables to constraints, d/n = gamma is fixed. On the other hand e.g. if n > d log(d), not unrealistic, then gamma = 0 and improvement stated before eq (4) no longer applies. - Refreshing sketches in each iteration seems rather expensive. log(1/eps) multiplying dominant terms in eq (18) in Section 6 seems worse than (17). E.g for eps < 1/d (high accuracy), (18) is always worse than (17). Could you discuss which theoretical results (do not) hold without re-sketching and empirically investigate how much of it could be skipped (i.e. sketch only once, or 10th iteration, etc)? Also, more recent (?) https://arxiv.org/abs/2006.05874 states "Surprisingly, refreshing embeddings does not improve on using a fixed embedding: it results in the same convergence rate in the Gaussian case [14, 15] and in a slower convergence rate in the SRHT case [15]." Could you make it clear what is the most up-to-date knowledge on refreshing sketches. It's fine if this paper's method and analysis is not the best for chronological and historical reasons, however please state of the art clear and provide a reference.

Correctness: The papers methodology is solid. Could you please release the code for reproducing the experiments?

Clarity: The paper is generally well written. Nevertheless it's quite technical, combining recent developments in advanced random matrix theory and iterative Hessian sketching and thus might require quite some dedication from non-experts.

Relation to Prior Work: The paper does a good job of describing related work and positioning its results. There exists quite some prior work showing that randomized Hadamard or Haar matrices have lower variance in dimensionality reduction. See e.g. The Unreasonable Effectiveness of Structured Random Orthogonal Embeddings Krzysztof Choromanski Mark Rowland Adrian Weller NeurIPS 2017 and later works from the same authors for many theoretical guarantees that non iid random matrices are more accurate than iid random matrices. Though these works don't consider iterative sketching and analyze other applications, it's probably worth citing.

Reproducibility: Yes

Additional Feedback: Theorem 3.1 : Theta_{1,h} is undefined here, only defined in subsequent Lemma 3.2. Section 5: Please include wall clock times, and time to solve it exactly and with (preconditioned) conjugate gradient [discussed in next section]? How was equation (2) with inverse of sketched Hessian solved? You could potentially also verify experimentally that theoretically best step size and (no) momentum hyper parameters work best.


Review 4

Summary and Contributions: The paper analyzes how the Gram matrix of a sketch approximates the identity (in a very specific fashion; see Lemma 3.2), and then uses this to analyze least-squares problems using the well-known Iterative Hessian Sketch framework. These results were known for Gaussian matrices, but not for partial Haar or subsampled randomized Hadamard transform (SRHT) matrices. This paper proves these results for Haar and SRHT matrices, though everything is asymptotic as d --> infty and assumes a limiting ratio of rows/columns. Notably, these results show that the Haar and SRHT matrices, which are partial orthogonal, outperform Gaussian, and that this explains practical observation. I am dubious about this being observed much in practice, and it would only show up as the size of the sketch increases (in the case of a square sketching matrix, there is a huge difference, since orthogonal sketching leads to no error; but of course, then there's no point in sketching in the first place in that regime). Because I was suspicious of the practical implications, and because the authors oversold their contributions a bit (all their claims of optimality and improvement are all in the limit as d--> infty), I was initially thinking this was a borderline submission, since it had correct results but I wasn't sure about the interest to the community. However, as I finished reading the paper, my opinion has improved, and I think this is quite a nice result. It is only asymptotic, but the numerical simulations actually show that it has some effect. The proof sketch provided in the main text is also quite nice. This does build on recent works like [14] and [8], but I think it stands on its own. In particular, I think the technique is nice, and the paper convinced me that this might be something I should look into myself. I keep from giving the paper a very high score because the practical effect is not that significant, it is only asymptotic, and the math is not a completely new technique (though it may be helping to bring a new technique into the sketching field.) == After reading the rebuttal == I'm still positive about the paper and encourage the other reviewers that it is worth accepting. However, some other reviewers raise good points, so please address those in the minor revision in the event that this is accepted.

Strengths: - Applies to least-squares (LS) problems, and LS problems are a benchmark system and used as proxies for harder problems - LS problems are also fundamental, like linear programming or sorting, so while the real-world numerical impact is debatable, the theoretical contribution is substantial - Numerics show that, qualitatively, the asymptotic results hold even for matrices when n ~ 1000. - The math seems quite good. I am not an expert on random matrix theory, so I did not give a detailed check of the proofs, but there were no obvious claims that seemed doubtful, and the proof sketch of the key lemma was good. May encourage others to use this kind of math for sketching. - The authors seemed well-aware of relevant sketching literature

Weaknesses: - Only applies to overdetermined least-squares (LS) problems, which is a bit uninteresting at this point. We can solve these quite well already. - Practical performance improvement by using orthogonal transforms is slight (only apparent as sketch becomes larger, where it is less useful; and in the regimes studied by numerical simulations, the decay factor rho was very small for all methods, so convergence was super fast for everything and the differences were not that great) - Everything is in the asymptotic regime, which is not obviously useful, and not very standard for sketching results - The language of the paper oversells things, since it keeps referring to things as "optimal" without qualifying that everything is asymptotic.

Correctness: Yes, as far as I can tell, everything is correct. There were some numerical simulations that were quite simple, and I don't see any issues with those (they showed what they were supposed to show, though it would have been nice to estimate a slope and get empirical values for rho to see how they matched the asymptotic ones)

Clarity: Yes, aside from some comments I have below about sequences, the math was clear, and the writing was done well. This paper was enjoyable to read.

Relation to Prior Work: The relation to prior work was made quite clear. In the sketching literature, this result is clearly novel. Whether such second-moment bounds were known in the random matrix literature is not clear to me.

Reproducibility: Yes

Additional Feedback: - The claims made in the contribution section (1.1) about "Haar embeddings have uniformly better performance than Gaussian ones" and about optimal parameters all feel oversold, since these are asymptotic results. Similarly when you refer to the "best known complexity", this is again asymptotic. Also, in Thm 3.1, interpreting rho_h as the optimal rate, again, this is only asymptotic, since you have liminf_t (lim_n( ...) ). If you had rho_h \le liminf_n( liminf_t(...)), it would strengthen your claim that this is optimal. - "It has been observed in several empirical contexts that random projections with i.i.d. entries degrade the performance of the approximate solution compared to orthogonal projections [16, 17, 9]. More recently, this observation has also found some theoretical support in limited contexts [8, 33]. Consequently, along with computational considerations, these results favor the SRHT over Gaussian projections." I think the first two sentences above are true, if you are thinking of truncated Haar vs Gaussian. But for small number of measurements (the useful regime), Haar and Gaussian are very similar, and in this regime, I am pretty sure Gaussians are better than SRHT for most tasks (like usual Johnson-Lindenstrauss guarantees or subspace embeddings). I haven't heard of people using SRHT because it is more *accurate* than Gaussian sketches. Your paper adds some support for this in the context of IHS, but I had never heard of this from other researchers. Furthermore, your numerical evidence in support of SRHT over Gaussian (in terms of accuracy) is quite slight. It's a clear difference (like having a small p-value) but the effect size is small. Your results on page 5 (for Haar) corroborate what I said: that the benefit of orthogonal occurs as xi --> 1 (i.e., as the sketching matrix becomes square). And of course, this makes perfect intuitive sense, since in this limit, the Gram matrix of any partial orthogonal sketch is the identity, while for the Gaussian it is the identity but only in expectation. So your result agrees with intuition, but it's not as clearly *useful* since sketching algorithms are interested with xi as small as possible, close to gamma. Other minor comments: - Line 50, why is || A(\tilde{x} - x^*) ||^2 a semi-norm? You've assumed that rank(A)=d. Also, remove the ^2 since you're referring to the norm, not the norm-squared. - Equation (2) has momentum, and I don't believe this was in the original IHS paper [23]; also, the original paper implicitly used a unit stepsize. This is probably in [14] (which I haven't read)? So please cite [14] for the algorithm associated with equation (2). - Line 68, for these sketch-to-precondition methods [4,24], I think it would be nice to add Saunder's LSRN paper as well. You could then explain that [4,24] use right pre-conditioning *and* SRHT (whereas LSRN used Gaussian). - Line 140, how can you define {X_{n1}, ..., X_{nI}} to be asymptotically free by saying X_{ni} has a limiting spectral distribution. You've fixed n, so how can you take a limit? Don't you mean you have a sequence (over n) of these families (over I)? Same in lines 145--147, you talk about A_n and B_n, but you must mean a sequence (A_n)_{n=1,2,...} and (B_n)_{n=1,2,...} - Ref [27] has typo in Hadamard; refs [34,35] are in italics.

[Author Response · NeurIPS 2020]

We thank the reviewers for their careful reading and constructive comments. We feel that the reviews are largely positive. In the remainder, we want to address some of the issues raised, and we will address them in detail in the revision. We will also release the code for the numerical experiments.

"Why would one care about the variable sketches, rather than the fixed sketch", "It's fine if this paper's method and analysis is not the best for chronological and historical reasons, however please state of the art clear and provide a reference." *This reference appeared after our work. It is based on the IHS with a fixed SRHT embedding, and their convergence rate appears currently the best known (in the asymptotic sense) for the SRHT. Differently, we emphasize that our analysis provides an **exact and closed-form formula of the convergence rate**. From a practical standpoint, their optimal fixed sketch algorithm critically relies on a momentum term, and this can be sensitive to noise and rounding. Without momentum, their fixed sketch algorithm has worse convergence rate than ours.*

"Did you experiment applying the truncated Walsh-Hadamard transform when using SRHT ?", "No numerical comparisons against state-of-the-art randomized methods are given.", "Could you [...] empirically investigate how [many refreshing steps] could be skipped?" *We'll include these additional experiments in the final version, and comparisons to state-of-the-art methods. In particular, our algorithm is faster than the pre-conditioned conjugate gradient method [24]. Further, it's more robust to noise in the gradients compared to the aforementioned fixed sketch algorithm. Although we do not have theoretical guarantees for skipping refreshing the embedding, we observe in practice that refreshing at each iteration can be omitted at the cost of convergence rate. Detailed numerical results will be included.*

"Practical performance improvement by using orthogonal transforms is slight", "The benefit of orthogonal occurs as $\xi \to 1$ [but we] are interested with $\xi$ as small as possible, close to $\gamma$.", "[The algorithm] might be slower [when] $\varepsilon$ cannot really be treated as a constant." *We emphasize one of our key findings, that is, the SRHT has the remarkable benefit of a fast projection method compared to Gaussian embeddings, along with always improving the convergence rate. For $\gamma = 0.1$, $\xi = 0.5$, the limiting convergence ratio between $\rho_h$ and $\rho_g$ is about 60%. So orthogonal still has benefit. Even with $\xi$ being close to $\gamma$, the ratio is still strictly less than one. Indeed, we improve theoretical time complexity when $\varepsilon$ is treated as a constant, which is a reasonable setting, and we will discuss more general cases in the final version.*

"Could you please precisely give the references claiming that $m \approx d \log(d)$ is prescribed for state-of-the-art algorithms and for which algorithm?" *Up to constant factors, the authors of [24] originally prescribed $m \geq d^2$ (see Lemma 1). Improved concentration bounds on the SRHT [27] can be used to improve this lower bound to $d \log d$. See also [6], Thm 3.1, where the bound $m \geq C \log d[\sqrt{d} + \sqrt{\log n}]^2$ is stated.*

"'For SRHT, we use the optimal step sizes'. I thought it was not proven to be optimal for SRHT. Did I miss something?" *We will make clearer that this step size is optimal conditional on $\beta = 0$.*

"A comment on how to deal in the case where n is not a power of 2." *One can use padding with zeros, which increases the value of $n$. This slightly increases the convergence rate. Or one can take a random subset of coordinates of SRHT, which empirically does not increase convergence rate, but is somewhat slower to compute.*

"How was equation (2) with inverse of sketched Hessian solved?" *In practice, the fastest method is to solve approximately the linear system with an iterative solver such as conjugate gradient.*

"Can this analysis be also extended to accommodate count-sketch or any other sparse sketching matrices?" *We rely on recent results in random matrix theory (RMT) which, to our knowledge, have not been derived for sparse embeddings. But there is recent work that analyzes both SRHT/Haar and some sparse embeddings in the same asymptotic framework, for PCA: arxiv.org/abs/2005.00511. It may work here but possibly with stronger assumptions on the data.*

"The SRHT in this paper incorporates an additional permutation.", "The optimality of IHS is only proved for Haar transform." *We do consider this additional permutation as we leverage recent results from the RMT. We cannot think of drawbacks of the extra permutation (computational or otherwise). For proving optimality for the SRHT, we would need results currently unknown in the RMT. Please see Remark A.1 for more details.*

"'SRHT [...] contains less randomness, but is more structured and faster to generate' than Haar matrix." *SRHT relies on a random permutation and $n$ sign-flips, while constructing Haar matrices usually needs order $n^2$ random Gaussian variables. Thanks to the matrix decomposition of the SRHT, multiplications are faster to perform via FFT.*

"Analysis is asymptotic and holds in the infinite limit only.", "... not very standard for sketching results", "Paper assumes that the ratio $d/n = \gamma$ is fixed.", "The language of the paper oversells things [...] without qualifying that everything is asymptotic." *The asymptotic framework is a good fit as one will only use sketching when the dataset is large. It also leads to clean theoretical results. Finite-sample results sometimes hide large constants. Our results are down to the constant and thus can be used easily by practitioners. The asymptotic result is also powerful enough to illustrate that Hadamard projection is superior to Gaussian projection. Moreover, the asymptotic results agree with simulations well with a few thousands samples. We will make our claims more precise and use asymptotically optimal wherever needed.*

[Meta-Review · NeurIPS 2020]

Four knowledgeable reviewers recommend accept, and I concur, in light of the contributions made: a sharp asymptotic analysis of the behavior of SRHT and Haar sketches, and the provision of optimal step sizes. Please revise your paper in accordance with the promises made in the author rebuttal, including the provision of numerical results on the effects of not refreshing the sketch at each iteration.